# Deletion of the *MAD2L1* spindle assembly checkpoint gene is tolerated in mouse models of acute T-cell lymphoma and hepatocellular carcinoma

Floris Foijer[1,2*†], Lee A Albacker[2†], Bjorn Bakker[1], Diana C Spierings[1], Ying Yue[2], Stephanie Z Xie[2‡], Stephanie Davis[2], Annegret Lutum-Jehle[2], Darin Takemoto[3], Brian Hare[3], Brinley Furey[3], Roderick T Bronson[4], Peter M Lansdorp[5], Allan Bradley[6], Peter K Sorger[2*]

[1]European Research Institute for the Biology of Ageing, University Medical Center Groningen, University of Groningen, Groningen, The Netherlands; [2]Department of Systems Biology, Harvard Medical School, Boston, United States; [3]Vertex Pharmaceuticals Incorporated, Cambridge, United States; [4]Rodent Histopathology Core, Dana-Farber/Harvard Cancer Center, Harvard Medical School, Boston, United States; [5]Terry Fox Laboratory, BC Cancer Agency, Vancouver, Canada; [6]Wellcome Trust Sanger Institute, Hinxton, United Kingdom

**\*For correspondence:** f.foijer@ umcg.nl (FF); peter_sorger@hms. harvard.edu (PKS)

[†]These authors contributed equally to this work

**Present address:** [‡]Princess Margaret and Toronto General Hospitals, University Health Network, Toronto, Canada

**Abstract** Chromosome instability (CIN) is deleterious to normal cells because of the burden of aneuploidy. However, most human solid tumors have an abnormal karyotype implying that gain and loss of chromosomes by cancer cells confers a selective advantage. CIN can be induced in the mouse by inactivating the spindle assembly checkpoint. This is lethal in the germline but we show here that adult T cells and hepatocytes can survive conditional inactivation of the Mad2l1 SAC gene and resulting CIN. This causes rapid onset of acute lymphoblastic leukemia (T-ALL) and progressive development of hepatocellular carcinoma (HCC), both lethal diseases. The resulting DNA copy number variation and patterns of chromosome loss and gain are tumor-type specific, suggesting differential selective pressures on the two tumor cell types.

## Introduction

Aneuploidy, the presence of an abnormal number of chromosomes in a cell, is a hallmark of solid tumors and in human cancers is frequently an indicator of poor prognosis. Genomic instability has the potential to promote loss of tumor suppressors and increases in oncogene copy number, thereby driving tumorigenesis. However, experiments in non-transformed cells show that chromosome imbalance imposes a physiological burden that reduces cell fitness (*Torres et al., 2007*; *Williams et al., 2008*; *Kops et al., 2004*; *Mao et al., 2003*). Primary murine embryonic fibroblasts (MEFs) engineered to carry an extra chromosome grow more slowly than wild-type cells and exhibit significant changes in metabolism. The same is true of cells from Down syndrome patients, which carry a supernumerary chromosome 21 (*Williams et al., 2008*; *Jones et al., 2010*). It remains poorly understood how the oncogenic effects of genomic instability as a driver of gene gain and loss, and the burden of aneuploidy in reducing fitness play out in real tumors. It has been suggested that tumors experience a burst of chromosome instability (CIN) leading to the emergence of clones with greater oncogenic potential but that CIN is then suppressed so that cancer cells maintain a relatively stable karyotype (*Lengauer et al., 1997*; *Wang et al., 2014*). This model of 'genome restabilization'

**eLife digest** An estimated 350 billion of the cells in the human body are dividing at any given moment. Every cell division requires the 46 chromosomes in the cell, which store the genetic information that the cell needs to survive, to be copied and distributed evenly between the two new cells. Sometimes mistakes in cell division can result in cells that have the wrong number of chromosomes – a state called aneuploidy.

Aneuploidy is rare in healthy cells but occurs in over 75% of cancers. It is the result of a process called chromosomal instability that often leads to the death of healthy cells. However, it is not well understood how aneuploidy affects how cancer cells develop or behave.

Mice are commonly used to investigate cancer because they have many genetic similarities with humans. To better understand the relationship between aneuploidy and cancer, Foijer, Albacker et al. engineered mice in which they could induce aneuploidy in liver cells and immune cells called T-cells. This modification accelerated the formation of liver cancer and lymphoma – a cancer of the immune system. The number of chromosomes in the cells of these cancers varied greatly, demonstrating that these cells experience constant chromosomal instability. Overall, this suggests that aneuploidy increases the likelihood of cancer developing.

The mouse cancer cells closely resemble their human counterparts, and so could potentially be used to test new cancer drugs. In the future, developing new therapies that selectively target aneuploid cells could result in cancer treatments that have fewer side effects than existing treatments.

is supported by statistical analysis of tumor karyotype across large numbers of human cancer genomes including identification of a common cancer karyotype, overlaid by tissue-specific differences, in genomic data from The Cancer Genome Atlas (*Davoli et al., 2013*). In addition, aneuploidy appears most common in areas of the genome having more oncogenes and tumor suppressors, implying positive selection. Conversely, however, many solid tumors are genetically heterogeneous, suggesting that CIN is not fully suppressed in growing tumors (*Nicholson and Cimini, 2013*).

Mouse models of CIN provide an excellent tool for studying the role of aneuploidy in tumorigenesis. One method to induce CIN is to disrupt the spindle assembly checkpoint (SAC). The SAC senses the presence of maloriented or detached kinetochores and blocks exit from mitosis until pairs of sister chromatids achieve the bipolar geometry that is uniquely compatible with normal chromosome disjunction (*Jallepalli and Lengauer, 2001*; *Taylor et al., 2004*; *Rieder et al., 1995*). Studying the SAC in mice is complicated by the fact that germline deletion of murine SAC genes is lethal by ~E10.5 (*Li et al., 2009*; *Dobles et al., 2000*; *Babu et al., 2003*; *Baker et al., 2004*; *García-Higuera et al., 2008*; *Iwanaga et al., 2007*; *Perera et al., 2007*; *Putkey et al., 2002*; *Wang et al., 2004*). As a result, most studies of SAC knockouts to date have employed heterozygous animals or hypomorphic alleles, resulting in weak and sporadic tumor development at long latencies (*Iwanaga et al., 2007*; *Burds et al., 2005*; *Dai et al., 2004*; *Michel et al., 2001*). We have previously shown that deletion of Mad2l1 (HUGO MD2L1; UniProt Q9Z1B5), an essential component of the SAC, is tolerated by murine interfollicular epidermal cells, which terminally differentiate to form the outer layers of the skin, but not by hair follicle bulge stem cells, a specialized self-renewing cell type required for hair follicle maintenance (*Foijer et al., 2013*). These findings support the idea that a functional SAC is required in cells undergoing repeated division, but not necessarily in differentiated cells with limited proliferative potential. The implications for cancer are unclear, since cancers grow from one or a small number of cells which must divide many times to create a macroscopic tumor.

In this paper we describe an analysis of tumorigenesis in mice carrying a conditional Mad2l1 deletion in a highly proliferative cell type (T-cells) and in a second cell type in which proliferation is induced by injury (hepatocytes). To tolerize cells to checkpoint loss we also introduced conditional deletions or mutations in Trp53. We found that T-cells and hepatocytes survive checkpoint loss in both the presence and absence of Trp53 mutation but that Trp53 mutations promote oncogenesis (*Jacks et al., 1994*; *Purdie et al., 1994*; *Donehower et al., 1992*). In T-cells, loss of Mad2l1 and Trp53 causes rapidly growing acute lymphoblastic leukemia (T-ALL) and in hepatocytes it causes

progressive disease that ends in lethal hepatocellular carcinoma (HCC). Single-cell sequencing shows that Mad2l1-null T-ALLs experienced an elevated rate of chromosome mis-segregation relative to normal T cells and murine T-cells in which the SAC is partially inactivated by truncation of Mps1 (Mps1 is another SAC component; (*Bakker et al., 2016*; *Foijer et al., 2014*). In contrast, when Mad2l1-null T-ALLs and HCCs were assayed at a population level using array-based comparative genomic hybridization (aCGH) recurrent and tissue-specific patterns of chromosome loss and gain were observed. The differences between single-cell and population-level measures of aneuploidy are most parsimoniously explained by postulating that Mad2l1-null tumors experience ongoing CIN but that specific aneuploid genomes predominate in a tumor as a result of tissue-specific selection.

## Results

### Conditional inactivation of Mad2l1 in thymus and liver

To cause CIN in a tissue-restricted fashion, we engineered a conditional flanked-by-LOX allele of Mad2l1 (Mad2l1$^f$). LoxP sites were inserted upstream of exon 2 and downstream of exon 5 so that Cre expression would result in deletion of ~90% of the Mad2l1 ORF (Mad2l1; *Figure 1A* and *Figure 1—figure supplement 1a*). Correct targeting of the construct in ES cells was confirmed by Southern blotting (*Figure 1—figure supplement 1b*). By crossing Mad2l1$^f$ and Lck-Cre transgenic animals we induced recombination of Mad2l1$^f$ in CD4$^-$ CD8$^-$ T cells (*Molina et al., 1992*) and by crossing with Alb-Cre carrying mice we induced Mad2l1$^f$ recombination in developing hepatocytes (*Weisend et al., 2009*), a post-mitotic cell type that normally exhibits polyloidy (*Duncan et al., 2010*). In both tissues, genotyping showed that Mad2l1 excision was efficient (*Figure 1B,C*). We generated a tumor-sensitized background by crossing Cre transgenic Mad2l1$^{f/f}$ and FLOX-Trp53 (Trp53$^f$) mice (*Jonkers et al., 2001*); Trp53 loss has been shown to promote survival of Mad2l1-deficient murine cells (*Burds et al., 2005*).

### Mad2l1 loss causes aggressive lymphoma and hepatocellular carcinoma in a Trp53 deficient background

Lck-Cre::Mad2l1$^{f/f}$::Trp53$^{+/+}$ mice did not experience malignancies within the first year of life (*Figure 1D*) and adult T cells from these animals developed normally, suggesting that T cells are tolerant of Mad2l1 loss. On a Trp53-heterozygous background (a Lck-Cre::Mad2l1$^{f/f}$::Trp53$^{f/+}$ genotype) loss of Mad2l1 resulted in death of ~50% of animals by ~8 mo. (from T-ALL, see below) whereas control animals heterozygous for a Trp53 deletion but carrying wild type Mad2l1 had the same lifespan as wild-type littermate controls (*Figure 1D*; blue lines, p<0.01). On a Trp53-homozygous deletion background, loss of Mad2l1 (i.e. an Lck-Cre::Mad2l1$^{f/f}$::Trp53$^{f/f}$ genotype) resulted in rapid disease progression with half of double mutant animals dead by ~4 mo. (*Figure 1D*; red line). Double mutant mice experienced a statistically significant acceleration in cancer development relative to mice homozygous for Trp53 deletion, which itself is known to be highly tumorigenic in thymocytes (*Figure 1D*; compare green and red lines, p<0.01).

Dyspnea (labored breathing) was observed shortly before the death of Mad2l1-mutant animals, consistent with thymic hypertrophy. Post-mortem analysis of tissues revealed a ~10–15 fold increase in the average mass of the thymus and 70% increase in the mass of the spleen (*Figure 1E,F*). Histological analysis of thymi demonstrated the presence of rapidly dividing blasts with irregular nuclei and abnormal DNA, consistent with lymphoma (*Figure 1G*, compare top and bottom panels). We conclude that Mad2l1 and Trp53 loss cooperate in oncogenic transformation of T-cells and that the combination is rapidly lethal.

To characterize cellular defects in double knockout mice, ~20 Lck-Cre::Mad2l1$^{f/f}$::Trp53$^{f/f}$ animals were euthanized prior to the appearance of dyspnea and thymocytes then analyzed. FACS showed that thymi from these animals contained numerous dividing and undifferentiated (blasting) cells in comparison to thymi from control animals (*Figure 1H*; blasts, red arrow, normal cells black arrow). In ~10% of Lck-Cre::Mad2l1$^{f/f}$::Trp53$^{f/f}$ animals thymi were macroscopically normal but they also contained an abnormal number of dividing and undifferentiated cells showing that this phenotype was fully penetrant. In most animals, the spleen also contained blasting cells and was enlarged, suggesting metastasis of T-cells to this organ (*Figure 1H* compare blasting population highlighted by red arrow in lower right panel with blasting cells in thymus). However, blasts were not observed in the

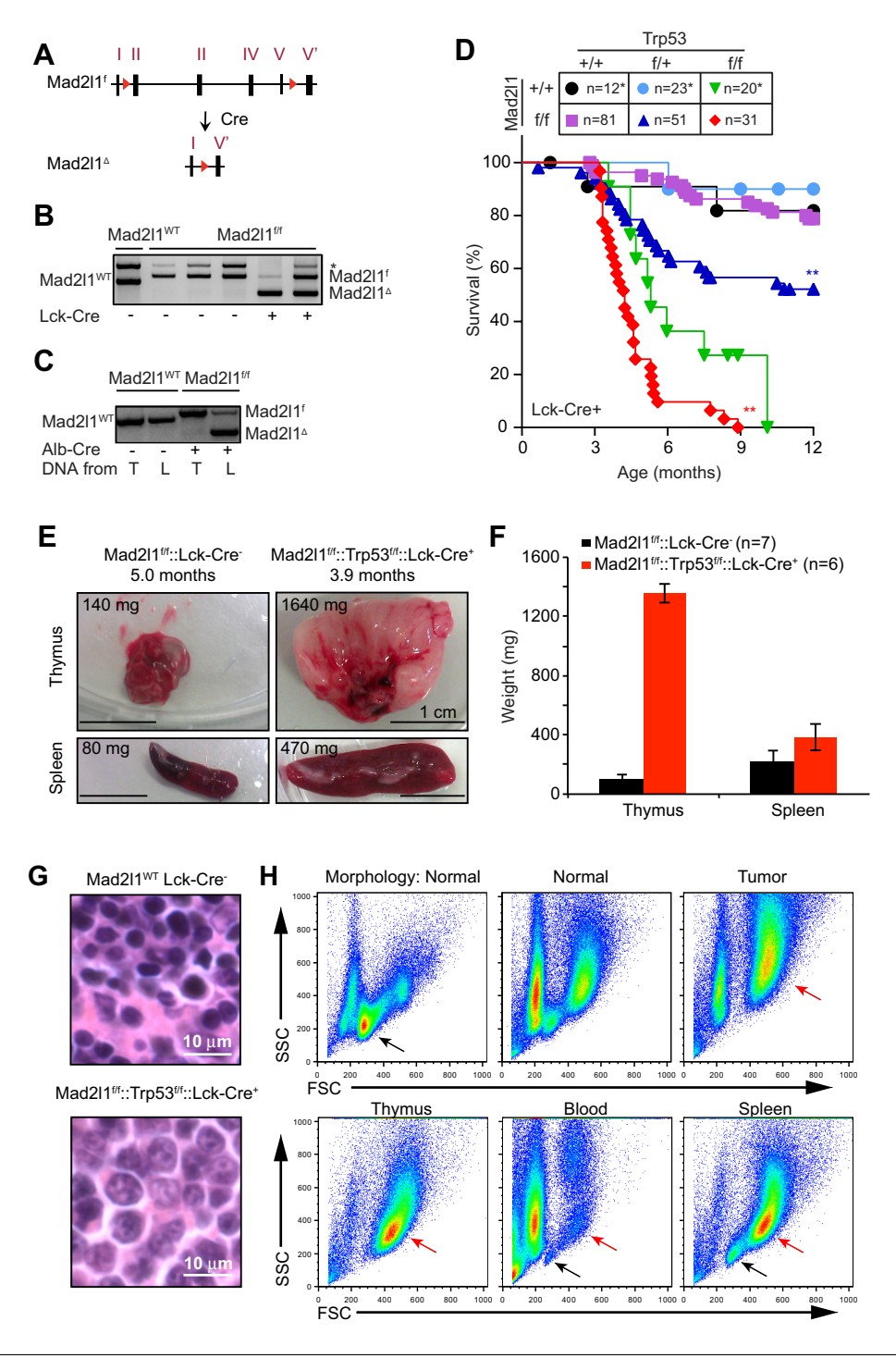

**Figure 1.** Tissue specific loss of Mad2l1 leads to T-cell acute lymphoblastic lymphoma in T cells in a permissive Trp53[null] background. (**A**) Schematic overview of the Mad2l1 conditional allele before and after Cre-mediated recombination. The red triangles refer to the loxP sites that surround exon 2 to exon 5; roman numerals refer to exons. (**B, C**) PCR for Mad2l1 genotypes and recombination of the *Mad2l1[f]* allele in (**B**) thymocytes and (**C**) liver tissue (L) or tail tissue (T). (**D**) Kaplan Meier plots showing survival of the indicated genotypes for *Lck-Cre::Mad2l1[f/f]::Trp53[f/f]* compared to control mice. Statistical tests for compared *Mad2l1[f/f]* and *Mad2l1[+/+]* having same *Trp53* genotype, \*\*p<0.01 (Mantel-Cox test). Control curves (*Lck-Cre::Trp53[f/f]* and *Lck-Cre*) were same animals as used in *Foijer et al. (2014)*. (**E**) Images showing enlarged thymus and spleen in a *Lck-Cre::Mad2l1[f/f]::Trp53[f/f]* mouse compared to a healthy control. (**F**) Average thymus and spleen weights for tumor-bearing *Lck-Cre::Mad2l1[f/f]::*

*Figure 1 continued on next page*

*Figure 1 continued*

*Trp53*[f/f] mice compared to unaffected control mice. (**G**) Representative H&E staining of control thymus (upper panel) and *Lck-Cre::Mad2l1*[f/f]*::Trp53*[f/f] acute T acute lymphoblastic lymphoma sample with staining indicating an undifferentiated cell state (lower panel). Scale bar 10 microns. (**H**) Forward and side scatter (FSC, SCC) plots for normal (appearing) thymuses and a T-ALL showing the emergence of a larger blasting population, before thymus size increased (upper panels). FSC and SCC plots for thymus, blood and spleen of a tumor-bearing mouse, showing blasting cells in thymus and spleen, but not blood (lower panels).

The following figure supplements are available for figure 1:

**Figure supplement 1.** Complete Targeting Vector and Generation of *Mad2l1*[f/f] Mice.

**Figure supplement 2.** Representative array CGH profiles for 3 *Lck-Cre::Mad2l1*[f/f]*::Trp53*[f/f] tumors showing clonal loss at the TCR loci on chromosomes 6 and 14 indicating tumor clonality.

---

peripheral blood (*Figure 1H* bottom center panel). These and related data show that the majority of *Lck-Cre::Mad2l1*[f/f]*::Trp53*[f/f] animals suffered from poorly differentiated CD4$^+$ and CD8$^+$ T-acute lymphoblastic lymphoma (T-ALL) while a subset suffer from more differentiated CD4$^+$ or CD8$^+$ T-ALL as described previously for Trp53$^{null}$ thymic lymphoma (*Donehower et al., 1995*). Array-based comparative genomic hybridization (CGH) analysis of TCR $\alpha$ and $\beta$ loci on chromosomes 14 and 6 revealed a single dominant rearranged TCR in each animal implying that T-ALLs were clonal (see *Figure 1— figure supplement 2*, array CGH data was deposited at GSE63686 in NCBI GEO). In sum, these data demonstrate synergy between loss of Mad2l1 and Trp53 in the transformation of T-cells to malignant T-ALL and show that tumors grow large enough to kill animals while remaining clonal at TCR loci. Mad2l1-null cells must therefore proliferate extensively subsequent to a tumor-initiating genetic event (a common characteristic of cancer).

The lifespan of animals carrying a conditional knockout of Mad2l1 in hepatocytes (*Alb-Cre:: Mad2l1*[f/f]*::Trp53*[+/+] mice) was unchanged relative to wild-type littermates (*Figure 2A*) but double mutant animals deleted for Mad2l1 and one or both Trp53 alleles (*Alb-Cre::Mad2l1*[f/f]*::Trp53*[f/+] and *Alb-Cre::Mad2l1*[f/f]*::Trp53*[f/f] mice) died significantly younger than littermate *Mad2l1*[+/+]*::Trp53*[+/+] controls (p<0.01 and p<0.001 respectively). In contrast, liver-specific deletion of one or both *Trp53* alleles in Mad2l1-wild type animals had no detectable impact on lifespan (*Figure 2A*; *Trp53*[f/+] blue lines, p<0.01; *Trp53*[f/f] red and green lines, p<0.001) consistent with previous data showing that Trp53 loss is only mildly oncogenic in hepatocytes (*Harvey et al., 1993*). Post-mortem analysis of *Alb-Cre::Mad2l1*[f/f]*::Trp53*[f/f] and *Alb-Cre::Mad2l1*[f/f]*::Trp53*[f/+] animals revealed the presence of one or more liver tumors per mouse. In many cases these tumors were so large and invasive that the tri-lobular structure of the liver was unrecognizable (*Figure 2B*).

In the case of single-mutant *Alb-Cre::Mad2l1*[f/f] animals, widespread liver damage and formation of regenerative nodules was evident by ~4 mo. of age. Regenerative nodules are non-neoplastic sites of liver proliferation and repair commonly present following liver damage. By 8–12 mo. of age benign hepatocellular adenoma (HCA) was evident in >50% of animals (~5% of wild type animals also had HCA, which is normal for this genetic background [*Leenders et al., 2008*]) and by 12–16 mo. half of mice had hepatocellular carcinoma (HCC; *Figure 2C*). Thus *Alb-Cre::Mad2l1*[f/f] mice experienced benign and malignant liver cancer at high penetrance. Deletion of *Trp53* on this background (*Alb-Cre::Mad2l1*[f/f]*::Trp53*[f/f] animals; *Figure 2D*) dramatically accelerated the onset of cancer, with 75% of 8–12 mo. old animals exhibiting HCC, occasionally in combination with HCA or cholangiocarcinoma (*Table 1*). Note that the reduction in the proportion of HCC *Alb-Cre::Mad2l1*[f/f] mice in the cohort older than 16 mo. arises simply because tumor-bearing animals die at an accelerated rate leaving behind disease-free animals. This is less obvious for *Alb-Cre::Mad2l1*[f/f]*::Trp53*[f/f] animals because nearly all of them eventually get HCC (so increasing death is matched by increasing disease prevalence).

People living in areas of the globe in which hepatotoxic agents are endemic often suffer from HCC that involves a dominant-negative Trp53-R249S mutation (*Yin et al., 1998*; *Hsu et al., 1991*; *Lee and Sabapathy, 2008*). When we crossed the murine analog of this mutation (Trp53-R246S) into Mad2l1-mutant animals (*Alb-Cre::Mad2l1*[f/f]*::Trp53*[R246S] mice; [*Yin et al., 1998*]) we observed early

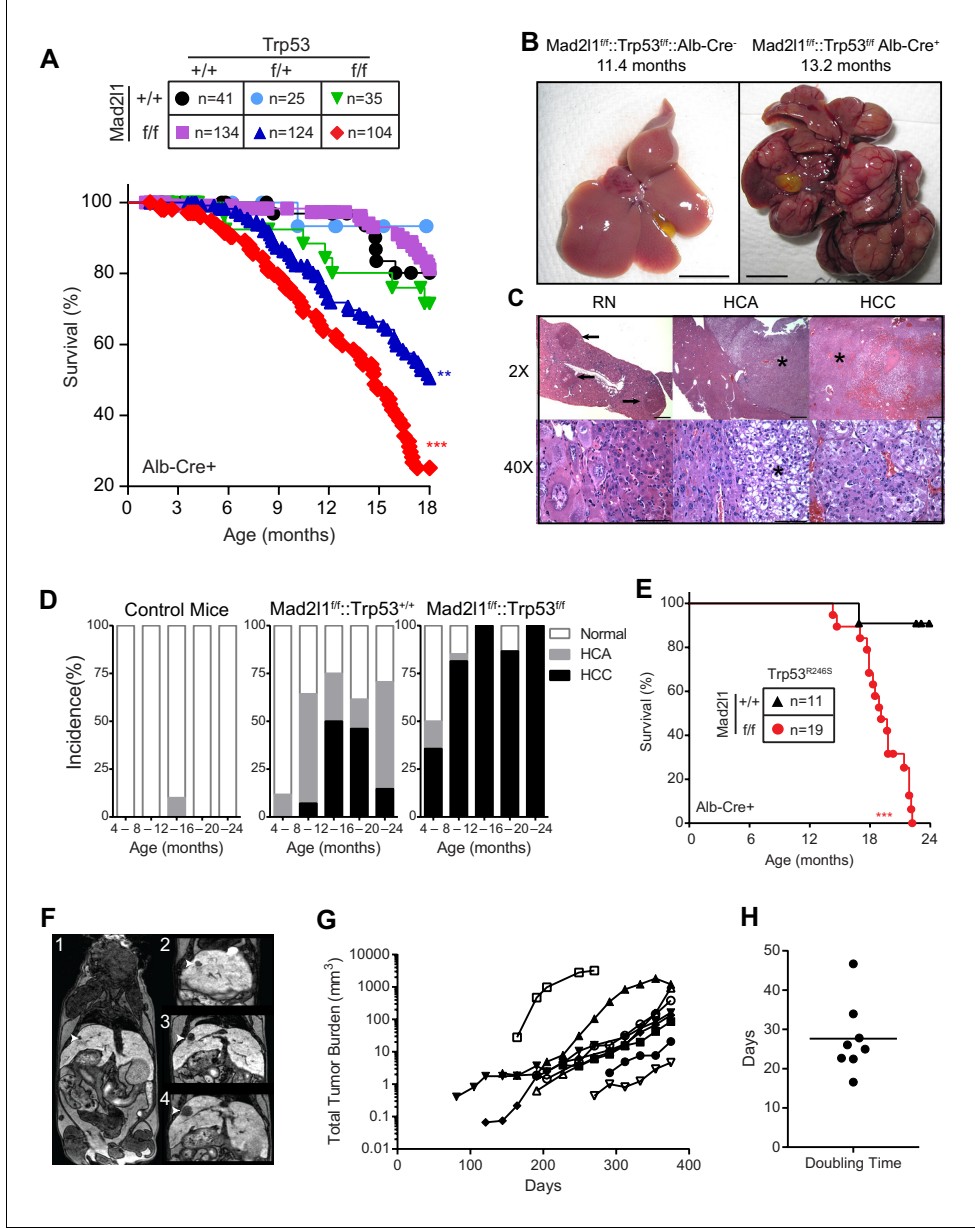

**Figure 2.** Loss of Mad2l1 in hepatocytes results in multifocal hepatocellular carcinoma. (**A**) Kaplan Meier plots showing survival of the indicated genotypes for *Alb-Cre::Mad2l1*$^{f/f}$*::Trp53*$^{f/f}$ compared to control mice. Statistical tests compared *Mad2l1*$^{f/f}$*::Trp53*$^{f/f}$ and *Mad2l1*$^{f/f}$*::Trp53*$^{f/+}$ to *Mad2l1*$^{+/+}$*::Trp53*$^{+/+}$ mice, **p<0.01, ***p<0.001 (Mantel-Cox test). (**B**) Images showing multifocal disease in a *Alb-Cre::Mad2l1*$^{f/f}$*::Trp53*$^{f/f}$ mouse compared to a healthy control. (**C**) Histological sections from *Alb-Cre::Mad2l1*$^{f/f}$*::Trp53*$^{+/+}$ mice. Left panels show regenerative nodules (RN) marked by black arrows in low magnification field and cells in one nodule at high magnification. Centre panels show an HCA marked by the asterisk. Right panels show an HCC marked by an asterisk (top) or the entire field (bottom). Scale bars in top fields 1 mm, bottom fields 0.1 mm. (**D**) Incidence of HCA and HCC in the livers of Control (*Mad2l1*$^{+/+}$*::Trp53*$^{+/+}$ or *Alb-Cre*$^{-}$), *Alb-Cre::Mad2l1*$^{f/f}$*::Trp53*$^{+/+}$, and *Alb-Cre::Mad2l1*$^{f/f}$*::Trp53*$^{f/f}$. (**E**) Kaplan Meier plot showing survival of *Alb-Cre::Mad2l1*$^{f/f}$*::Trp53*$^{R246S}$ mice compared to control. ***p<0.001 (Mantel-Cox test). (**F**) Representative MRI images of an *Alb-Cre::Mad2l1*$^{f/f}$*::Trp53*$^{f/f}$ with a tumor (white arrow) over time in weeks. EOVIST is used as a contrast agent and tumors exclude the reagent and are dark. (**G**) Volumetric measurements of tumors over time. Each symbol represents a different mouse. (**H**) Doubling time as determined by semi-log regression of data in (**G**).

The following figure supplement is available for figure 2:

*Figure 2 continued on next page*

*Figure 2 continued*

**Figure supplement 1.** Characterizing aneuploid T-ALLs and HCCs.

death and extensive HCC (*Alb-Cre::Mad2l1^{f/f}::Trp53^{R246S}* vs. *Alb-Cre:Trp53^{R246S}* littermates; p<0.001; compare *Figure 2A and E*, red lines). Such animals therefore recapitulate a known feature of human disease. We conclude that Mad2l1 deletion in hepatocytes is sufficient to cause HCC but that tumor formation is significantly accelerated by deletion of Trp53 or introduction of a mouse analog of a Trp53 mutation commonly observed in human liver cancer.

To analyze disease progression in *Alb-Cre::Mad2l1^{f/f}::Trp53^{f/f}* animals we used magnetic resonance imaging with EOVIST as a contrast reagent. EOVIST-excluding regions were confirmed to be tumors by fixing livers immediately after MRI followed by serial sectioning and H&E histology of the same region of the tissue (*Figure 2—figure supplement 1*). In nine animals examined (a total of 32 tumors) we observed that tumor volume increased approximately exponentially (*Figure 2F,G*) with an average doubling time of ~28 days (*Figure 2H*). Moreover, imaging revealed that the number of tumors in each animal increased with age so that by 12 mo. an average of 3 morphologically distinct tumors were present in each animal (range 1–7; n = 9). We conclude that liver cancer induced by loss of Mad2l1 and Trp53 results in progressive multi-focal cancer and that tumors grow exponentially once established. This resembles human HCC, which is also multi-focal and progressive.

## Mad2l1 deletion abrogates the SAC in vivo and in vitro and yields SAC-deficient tumors

Although we have previously shown that partially inactivating the SAC via truncation of Mps1 in T-cells accelerates Trp53-induced lymphomagenesis (*Foijer et al., 2014*), the observation that Mad2l1-null T-ALLs and HCCs can grow rapidly is surprising. Germline deletion of murine Mad2l1 is

**Table 1.** Incidence of HCA, HCC, and CC (cholangiocarcinoma) in *Alb-Cre::Mad2l1::Trp53* mice. The HCA/CC column describes the number of mice with HCC that also have HCA or CC.

| | 4–8 (Mo) | | | | 8–12 (Mo) | | | 12–16 (Mo) | | | 16–20 (Mo) | | | 20–24 (Mo) | | | |
|---|---|---|---|---|---|---|---|---|---|---|---|---|---|---|---|---|---|
| | HCA | HCC | HCA/CC | CC | HCA | HCC | HCA/CC | HCA | HCC | HCA/CC | HCA | HCC | HCA/CC | HCA | HCC | HCA/CC | CC |
| *Mad2l1^{f/f}: :Trp53^{+/+}* | 2/17 | 0/17 | / / | 0/17 | 8/14 | 1/14 | / / | 1/4 | 2/4 | / | 2/13 | 6/13 | 5/6 / | 19/34 | 5/34 | 3/5 | 0/34 |
| *Mad2l1^{f/f}: :Trp53^{f/f}* | 2/14 | 5/14 | 2/5 2/5 | 1/14 | 1/27 | 22/27 | 8/22 / | 0/9 | 9/9 | 4/9 1/9 | 0/15 | 13/15 | 6/13 / | 0/4 | 4/4 | 1/4 2/4 | 0/4 |
| *Mad2l1^{f/f}: :Trp53^{f/+}* | 2/12 | 2/12 | 1/2 / | 1/12 | 2/17 | 3/17 | 1/3 2/3 | 0/8 | 5/8 | 3/5 / | 11/25 | 12/25 | 6/12 2/12 | 3/10 | 5/10 | 3/5 | 0/10 |
| *Mad2l1^{f/+}: :Trp53^{f/f}* | 0/10 | 0/10 | / / | 0/10 | 0/10 | 0/10 | / / | 0/1 | 0/1 | / | 1/11 | 2/11 | / 1/2 | 2/13 | 4/13 | 1/4 2/4 | 1/13 |
| *Mad2l1^{f/+}: :Trp53^{f/+}* | 0/13 | 0/13 | / / | 0/13 | 0/7 | 0/7 | / / | 0/3 | 0/3 | / | 2/8 | 0/8 | / / | 0/8 | 2/8 | / | 0/8 |
| *Mad2l1^{f/+}: :Trp53^{+/+}* | 0/15 | 0/15 | / / | 0/15 | 0/6 | 0/6 | / / | 0/3 | 0/3 | / | 1/8 | 0/8 | / / | 2/13 | 2/13 | 1/2 | 0/13 |
| *Mad2l1^{+/+}: :Trp53^{f/f}* | 0/2 | 0/2 | / / | 0/2 | 0/4 | 0/4 | / / | 0/2 | 1/2 | 1/2 1/2 | 3/10 | 2/10 | 1/2 / | 0/1 | 1/1 | / | 0/1 |
| *Mad2l1^{+/+}: :Trp53^{f/+}* | NA | NA | | NA | NA | NA | | 0/2 | 0/2 | / | 0/4 | 0/4 | / | 0/1 | 0/1 | / | 0/1 |
| *Mad2l1^{+/+}: :Trp53^{+/+}* | 0/3 | 0/3 | / | 0/3 | 0/2 | 0/2 | / | 0/3 | 0/3 | / | 0/3 | 0/3 | / | 0/4 | 0/4 | / | |
| *Alb-cre^{−}* | 0/6 | 0/6 | / | 0/6 | 0/9 | 0/9 | / | 1/7 | 0/7 | / | NA | NA | | NA | NA | | NA |

lethal and RNAi-mediated depletion of Mad2l1 in cancer cells results in mitotic catastrophe and cell death within approximately six cell doublings (*Kops et al., 2004*; *Dobles et al., 2000*). To establish that the Mad2l1 locus was in fact lost in tumor cells and that no RNA or protein was expressed, we analyzed DNA structure by genomic PCR (*Figure 3A*) and aCGH (*Figure 3B*), mRNA levels using qPCR (*Figure 3C*), and protein levels by Western blotting of tumor samples (*Figure 3D*). In all but one T-ALL tumor from *Lck-Cre::Mad2l1^{f/f}::Trp53^{f/f}* animals (tumor 33, in which switching was incomplete and protein present) *Mad2l1* DNA, mRNA, and protein were below the level of detection. In the case of HCCs from *Alb-Cre::Mad2l1^{f/f}::Trp53^{f/f}* animals, PCR revealed bands corresponding to both recombined and unrecombined Mad2l1 (*Mad2l1^Δ* and *Mad2l1^f*). Probe values by aCGH were generally intermediate between probe values for wild-type cells and Mad2l1-null T-ALLs (compare *Figure 3B* to *Figure 3E*). The presence in HCC DNA from recombined and unrecombined Mad2l1^f loci is expected since liver comprises multiple cell types and tumors contain high levels of infiltrating Mad2l1-proficient immune cells. Mad2l1-null HCCs were invasive so resected tumors were invariably contaminated with surrounding normal tissue (including tissue in which recombination might have been incomplete). It is difficult to estimate the contribution of such cells to PCR signals, but in a subset of HCCs, *Mad2l1^Δ* was the dominant PCR product (tumors in lanes 6 and 11 in *Figure 3F*). It is also possible that cells heterozygous for Mad2l1 deletion can contribute to tumorigenesis. However, germline Mad2l1 heterozygosity did not cause HCC either alone or in combination with Trp53 deletion and we therefore find this explanation less likely (*Table 1*). Overall we conclude that Mad2l1 protein and mRNA are present in amounts below the level of detection in virtually all T-ALLs and

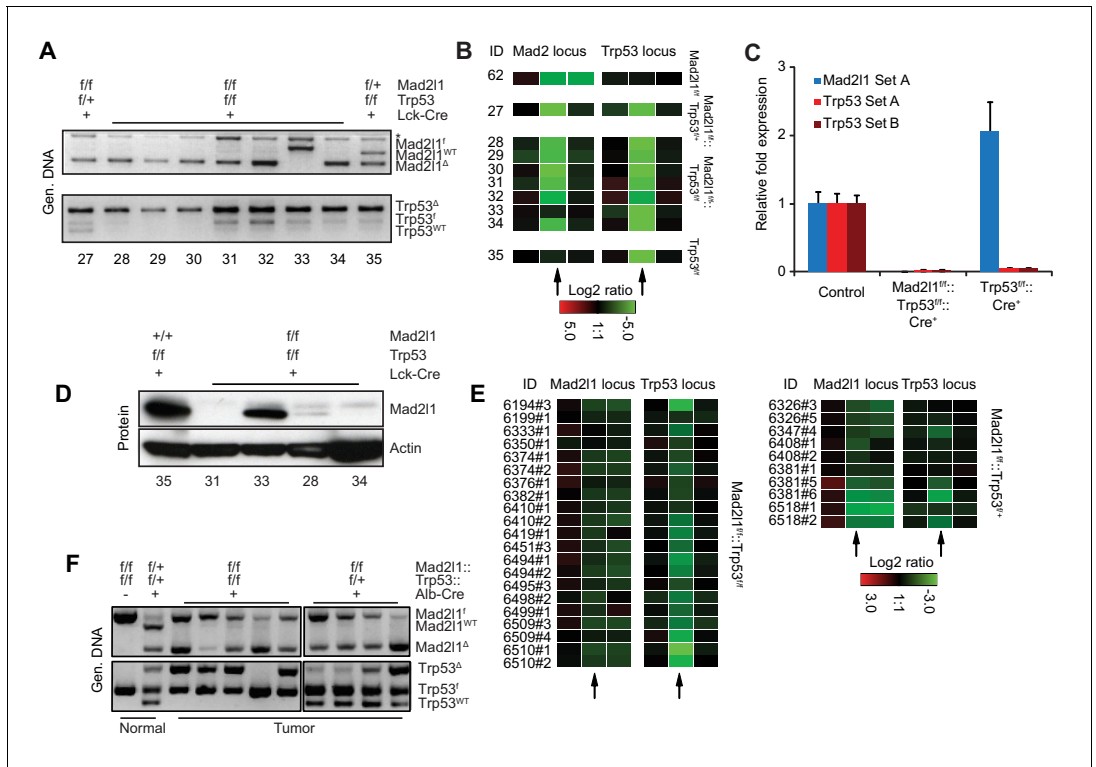

**Figure 3.** Excision of *Mad2l1* DNA and loss of SAC function in normal and tumor cells. (A) Recombination efficiency of *Mad2l1^f* and *Trp53^f* alleles in T-ALL samples as measured by genomic PCR. Numbers refer to tumor ID. (B) Recombination efficiency at *Mad2l1* and *Trp53* loci in T-ALL or samples as measured by array CGH. Each rectangle represents a single aCGH probe value, three probes values are shown per conditional gene: one probe recognizing the Mad2l1 or Trp53 deleted fragment (middle) and two probes flanking the 5′ and 3′ sides of the deleted region. (C) Quantitative PCR showing complete deletion of *Mad2l1* (probe A) and *Trp53* (probes A and B) in T-ALL samples. Error bars show SEM for six (*Mad2l1, Trp53*) and three (*Trp53*) tumors (biological replicates). (D) Western blots showing loss of Mad2l1 expression in T-ALL samples. (E) Recombination efficiency at *Mad2l1* and *Trp53* loci in HCC samples as measured by array CGH. (F) Genomic PCR of tumor tissue for WT, FLOX, and recombined alleles of *Mad2l1* and *Trp53*. Black vertical line shows where an empty lane was removed.

that in some HCCs, the extent of recombination is sufficiently high that some, and perhaps all transformed hepatocytes lack a functional Mad2l1 gene.

To demonstrate that recombination in *Mad2l1^{f/f}* cells abrogates checkpoint control we established *Mad2l1^{f/f}* mouse embryonic fibroblasts (MEFs) from E12.5-E13.5 embryos and infected cells with a retrovirus that expressed doxycycline- (Dox) inducible Cre (Dox-Cre; [*Foijer et al., 2014*]). Addition of Dox to *Mad2l1^{f/f}* MEFs that express Dox-Cre resulted in efficient excision of Mad2l1 as judged by PCR genotyping and Western blotting for Mad2l1 protein (*Figure 4A*). The resulting Mad2l1-null MEFs could be passaged multiple times in low oxygen conditions. Exposure of wild-type MEFs to nocodazole for 6 hr. increased mitotic index ~20 fold, reflecting checkpoint-dependent detection of microtubule depolymerization and imposition of cell cycle arrest. In contrast, when *Mad2l1^{Δ/Δ}* MEFS were exposed to nocodazole, only a modest increase in mitotic index was observed, consistent with a loss of SAC–mediated mitotic delay (*Figure 4B* and *Video 1*).

When wild type and *Lck-Cre::Mad2l1^{f/f}::Trp53^{+/+}* mice were injected with the microtubule stabilizing drug Paclitaxel, the fraction of phospho-H3 positive *CD4^+CD8^+* thymocytes (the most rapidly dividing thymic population) increased 6-fold in *Mad2l1*-sufficient animals but only 2.5-fold in Mad2l1-null animals (*Figure 4C*). Thus, the mitotic checkpoint is functionally impaired in *Mad2l1^{Δ/Δ}* thymocytes in vivo, consistent with loss of expression of a protein required for the spindle assembly checkpoint protein. We conclude that loss of SAC function is compatible with rapid growth of both liquid and solid tumors in the mouse.

The partial mitotic arrest observed in *CD4^+CD8^+* thymocytes from *Lck-Cre::Mad2l1^{f/f}::Trp53^{+/+}* animals might reflect the presence of Mad2l1-proficient T-cells in circulation combined with the inability of paclitaxel to impose a strong arrest on wild-type murine thymocytes (so the positive signal is not very high). Alternatively, it is possible that *Mad2l1^{Δ/Δ}* thymocytes are partially responsive to microtubule depolymerization and retain some SAC function. We are as-yet unable to distinguish between these possibilities. Despite attempts to raise suitable anti-mouse Mad2l1 antibodies or purchase them from commercial suppliers, we have been unable to perform sufficiently good immunofluorescence microscopy against murine Mad2l1 to determine whether the subset of thymocytes that arrest in the presence of paclitaxel express Mad2l1. Further development of single-cell methods will be needed to resolve this issue.

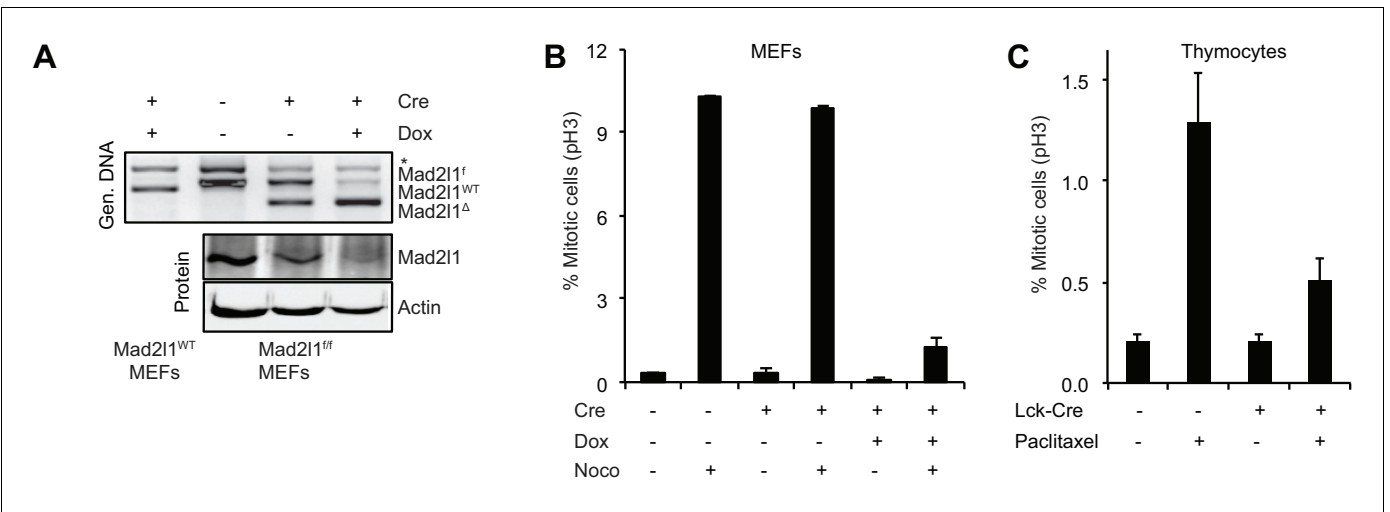

**Figure 4.** Mad2l1 inactivation in mouse embryonic fibroblasts fully alleviates the spindle assembly checkpoint. (A) Recombination efficiency measured by genomic PCR (top) and Western blot (bottom) showing partial to complete Mad2l1 deletion in MEFs following retroviral doxycycline-inducible Cre. Actin serves as loading control. (B) Average phospho-Histone H3 staining of dox-inducible Cre-transduced MEFs following 6 hr of nocodazole treatment. Error bars show the SEM of at least two biological replicates. (C) Average mitotic index of thymocytes isolated from Paclitaxel or control-injected mice 4–6 hr post-treatment. Error bars show SEM of at least four biological replicates.

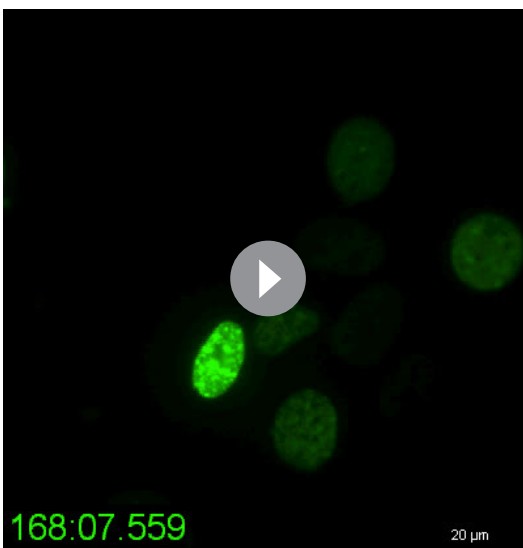

**Video 1.** SAC loss in dox-inducible Cre-transduced *Mad2l1^f/f* MEFs. Video shows the instant mitotic exit of dox-inducible Cre-transduced *Mad2l1^f/f* MEFs in the presence of nocodazole, which indicates loss of SAC function. Chromatin is labeled with H2B-GFP.

## Recurrent but tissue-specific CNVs and patterns of whole chromosomes aneuploidy

To study changes in the genome accompanying Mad2l1 deletion we performed genome-wide aCGH and microarray transcriptome analysis of T-ALL and HCCs (see Materials and methods, NCBI GEO, GSE63689). T-ALLs had a median of 3 whole chromosome loss or gain events per tumor, whereas HCCs had a median of 2 (*Figure 5A*, p>0.05). Loss of Chr13 was frequent in both tumor types (*Figure 5B,C*; green point in *Figure 5C*) as was gain of Chr15 (*Figure 5B,C*; red point in *Figure 5C*) which was also the most common whole-chromosome aneuploidy overall. Chromosomes also exhibited tissue-specific patterns of loss and gain: T-ALLs frequently gained Chr4 and Chr12 while HCCs lost these chromosomes (*Figure 5B,C*; Chr4: p<0.001, Chr12: p<0.01, blue points in 4E). In contrast, Chr18 exhibited preferential gain in HCCs but loss in T-ALLs (*Figure 5B,C*; p<0.001, blue points in *Figure 5C*; aCGH of individual tumors is shown in *Figure 5—figure supplement 1A* for T-ALLs and *Figure 5—figure supplement 1B* for HCCs).

One known effect of aneuploidy is to alter mRNA expression through gene dosage. On a per-chromosome basis we observed statistically significant correlation between chromosome ploidy and levels of mRNA expression from genes expressed on that chromosome (*Figure 5D,E*). However, the genes that were differentially expressed in HCC and T-ALL were not the same. This was true even for genes expressed on Chr15, which was gained in both HCC and T-ALL (*Figure 5F,G*). When we used Webgestalt (*Zhang et al., 2005*) to determine whether the same pathways were affected in T-ALLs and HCC, the only common denominators were metabolic pathways, in agreement with earlier findings showing that aneuploidy disrupts metabolism in multiple organisms (NCBI GEO GSE63689) (*Williams et al., 2008*; *Torres et al., 2007*; *Foijer et al., 2014*). We conclude that deletion of Mad2l1 results in loss and gain of whole chromosomes in both HCC and T-ALL but that the pattern of gain and loss is tissue specific, most likely due to changes in gene expression that are themselves tissue-specific. Moreover, even when the same chromosome is gained in HCC and T-ALL, the genes that exhibit differential expression are not the same.

For both T-ALL and HCC aCGH revealed recurrent copy number variants (CNVs), focal changes in chromosome structure, some of which included known oncogenes and tumor suppressors. We found that ~30% of T-ALLs carried a 1–8 gene deletion spanning *Pten* (*Figure 5—figure supplement 2A*), a negative regulator of the PI3 kinase and 5–10% of HCCs carried an amplification in *Met*, a known oncogene in human HCC (*Zender et al., 2006*). In the latter case, we confirmed that Met RNA and protein were over-expressed (*Figure 5—figure supplement 2B–D*). We detected no consistently amplified or deleted genes common to both thymus and liver tumors: three possible candidates uncovered by aCGH proved to be artifacts arising from our use of a mixed 129 x C57BL/6 background (*Boyden and Dietrich, 2006*) (*Figure 5H*). Moreover, the overall structure and pattern of gene amplifications and deletions appeared to differ in the two tumor types. Focal CNVs were more frequent in T-ALL than HCC: T-ALLs carried a median of 29 CNVs per tumor with an average size of ~40 genes per CNV whereas HCCs carried ~20 CNVs per tumor with an average size of ~2–3 genes (both differences were statistically significant, ~p< 0.05 and p<0.01 respectively; *Figure 5I,J*). This implies either differential selection for CNVs in T cells and hepatocytes or differences in chromosome breakage and rejoining as a consequence of SAC loss. We note a potential complication in the interpretation of this data. Whereas T-ALLs appear to be clonal based on TCR sequence HCCs are

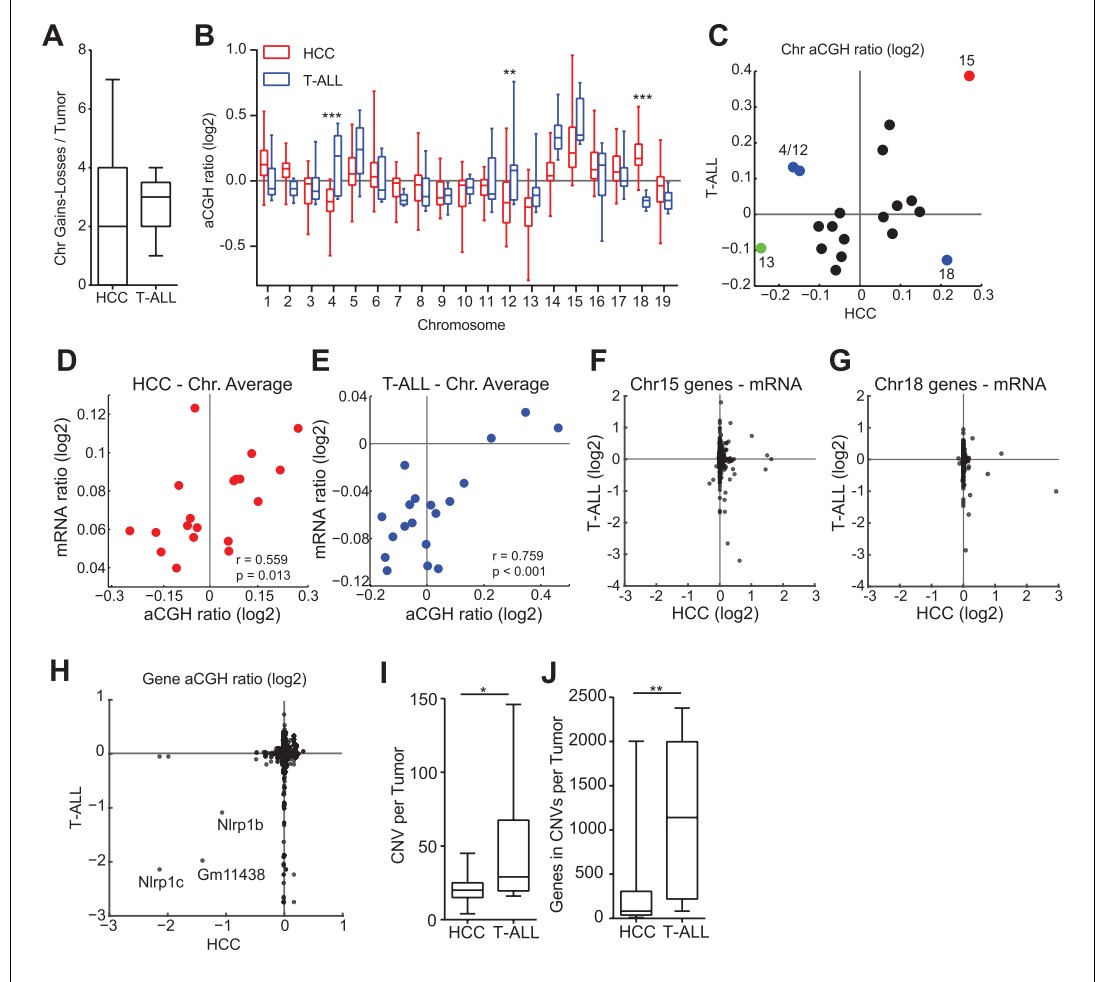

**Figure 5.** Aneuploidy events are a recurrent genetic lesion in T-ALLs and HCCs. (**A**) Box and whiskers plot of whole chromosome gain-loss events in tumors, scored by averaging the log2 ratio of aCGH fluorescence from tumor over normal (aCGH ratio) for each chromosome with a ±0.3 cut-off. Line – median, Box – interquartile range, Whisker – range. (**B**) Box and whiskers plot of the aCGH ratio for each chromosome. Two-way ANOVA, Bonferroni post-test **p<0.01, ***p<0.001 (**C**) Scatter plot showing average chromosome aCGH ratio of HCC and thymus tumors. Gain of Chr15 (red), loss of Chr13 (green), and tissue specific gain or loss of Chr4, Chr12, and Chr18 (blue). (**D, E**) Scatter plots of average chromosome aCGH ratio plotted against average mRNA ratio for (**D**) HCC and (**E**) T-ALL. r is the Pearson Correlation with indicated P value. (**F, G**) Expression analysis of genes on chromosome 15 (**F**) and chromosome 18 (**G**) comparing HCC and T-ALL samples. (**H**) Scatter plot showing chromosome normalized aCGH ratio for every gene in HCC and T-ALL. The three listed genes are likely hybridization artifacts due to a mixed 129/C57BL6 background. (**I, J**) Number of focally amplified or deleted regions per tumor (**I**) and the number of genes amplified or deleted per tumor (**J**) scored with ±0.3 cut-off for the chromosome normalized aCGH ratio. *p<0.05, **p<0.01, Mann-Whitney Test comparing T-ALL and HCC.

The following figure supplements are available for figure 5:

**Figure supplement 1.** Copy number changes in T-ALL and HCC as assessed by aCGH.

**Figure supplement 2.** T-ALL and HCC specific CNVs.

likely to be polyclonal even after physical dissociation of tumor masses. The degree of clonality might affect our assessment of CNV characteristics by aCGH (which averages across all cells in the sample). Analysis of additional tumors using single-cell methods will therefore be required to confirm the observation that *Mad2l1* deletion generates CNVs with different sizes in different tissues.

# Ongoing CIN results in tumor progression and intratumor karyotype heterogeneity

Mice lacking Mad2l1 in hepatocytes exhibited a characteristic progression from regeneration nodules to HCA and then HCC. To determine if HCA actually gives rise to HCC in *Alb-Cre::Mad2l1^{f/f}:: Trp53^{f/f}* and *Alb-Cre::Mad2l1^{f/f}::Trp53^{f/+}* animals, gain and loss of chromosomes was profiled by aCGH in HCA, HCC, and unaffected liver (see Materials and methods, NCBI GEO GSE63689 and GSE63100). Chromosomes 15, 16, and 18 were frequently gained in HCC (as described above) and also in HCA, although signals were weaker in HCAs implying that only a subset of cells had undergone chromosome gain events (*Figure 6A*). Overall, HCA appeared more aneuploid than normal tissue and HCC more aneuploid than HCA (*Figure 6B*). We assayed the relatedness of HCA and HCC by sequencing tumors from eight animals carrying a macroscopic HCA and one HCC and an additional mouse carrying one HCA and two HCCs (see Materials and methods, data deposited at Sequence Read Archive accession number SRA191233). CNVs in the benign and malignant tumors were compared pairwise using the Jaccard Index, which scores similarity while accounting for differences in the total number of alterations in each tumor (*Lohr et al., 2014*). To assess statistical significance, Jaccard Indexes were transformed into Z-scores. A pair of tumors was considered strongly related (at 95% confidence) if the Z-score was >1.96 in a two-way test (comparing the HCA to the HCC and the HCC to the HCA), while a pair was considered weakly related if only one side of the Z-score was >1.96. We identified two pairs of HCA and HCC tumors that were strongly related and found that the two HCCs that arose in a single mouse were also strongly related (relative to tumors from different animals; *Table 2*). In three animals HCA and HCCs were weakly related and in four animals we did not detect significant similarity (*Table 2*). In aggregate, tumor pairs from the same mouse were significantly more similar to each other than to tumors from other mice, but only 10–20% of CNVs overlapped among benign and malignant tumors from the same animal. We conclude that HCC can indeed evolve from HCA in our mouse model but that HCA may not be a necessary precursor to HCC. The frequency of progression is difficult to judge: when HCA and HCC appear unrelated, it is possible that a related HCA was present at an earlier time. Nonetheless, our data clearly demonstrate the possibility of evolution from HCA to HCC in a single animal and of a single HCC into a macroscopically distinct HCC in the same organ. Both progression and clonal evolution are accompanied by ongoing gain and loss of whole chromosomes and CNVs.

To measure intratumor karyotype heterogeneity in Mad2l1-null tumors directly we performed single-cell sequencing from three animals with T-ALL and wild-type control. Single cell suspensions of T-ALL and control thymocytes (~45 cells per animal) were separated by flow cytometry into 96 well plates followed by barcoding and low-coverage next-generation sequencing (*van den Bos et al., 2016*). As expected, no chromosomal abnormalities were observed in T-cells from wild-type thymi (*Figure 6C*) but extensive aneuploidy was evident in 3 T-ALL samples (*Figure 6D–F*). Most cells from all three T-ALLs subjected to single-cell analysis had gained Chr. 14 and 15 (*Figure 6D–F*) and in one of two tumors examined, a preponderance of cells had gained Chr. 1, 2, 4, 5, and 17 (*Figure 6F*). In contrast, changes in the ploidy of other chromosomes appeared random: > 70% of the cells in a single T-ALL had unique karyotypes. Intratumor differences in karyotype were further evidenced by high heterogeneity scores for all three T-ALLs (*Figure 6—figure supplement 1*). These scores were also higher than those of T-ALLs expressing truncated Mps1 (*Foijer et al., 2014*; *Bakker et al., 2016*). Because T-ALLs were clonal based on TCR loci, the most likely explanation for intratumor karyotypic heterogeneity is ongoing chromosome loss and gain, presumably due to loss of SAC function.

Human cancers in the Cancer Genome Atlas (TCGA) have previously been reported to have an abnormal pan-cancer karyotype involving Chrs. 7, 20 (which are frequently gained) and Chrs. 10, 13, 22 (which are frequently lost) (*Davoli et al., 2013*; *Nicholson and Cimini, 2013*) as well as tissue-specific changes in chromosome ploidy. To compare these findings with our results, we reanalyzed TGCA data for epithelial cancers such as HCC and breast cancer and also non-epithelial cancers such as glioma and glioblastoma multiforrme (*Figure 7*). In human HCCs, chromosomes Chr1q and Chr8q are commonly gained and Chr8p lost (*Figure 7*; top panel), an event that has been associated with worse clinical outcomes (*Emi et al., 1993*). Human 8q is related to murine Chr. 15, which we find frequently gained in Mad2l1-null HCCs (*Hertz et al., 2008*; *Guan et al., 2000*). Similar karyotypic abnormalities are found in HCC and breast cancer (*Figure 7*, top panel), consistent with the

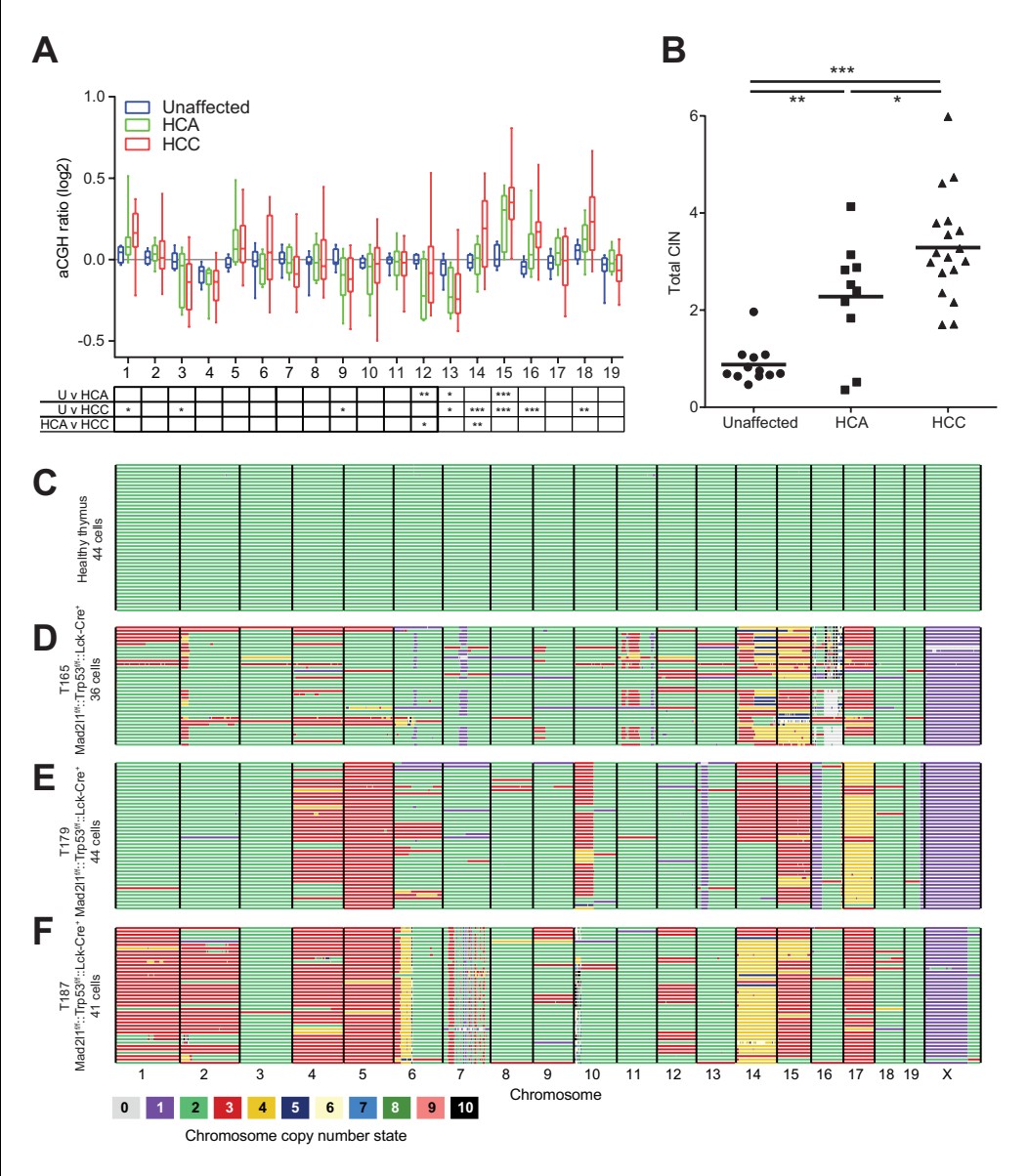

**Figure 6.** Mad2l1 deficiency results in clonal abnormalities despite ongoing chromosomal instability in murine T-ALL. (**A**) Box and whiskers plot for each chromosome of unaffected liver (blue n = 12), HCA (green n = 10), and HCC (red n = 18) from *Alb-Cre::Mad2l1^{f/f}::Alb-Cre* mice with mixed Trp53 genotypes. Statistical significance assessed by Two-way ANOVA and Tukey multiple comparison test with comparison between each group shown in the table below, \*p<0.05, \*\*p<0.01, \*\*\*p<0.001. (**B**) Sum of the absolute value of the aCGH ratio for each chromosome. Statistical significance assessed by One-way ANOVA and Tukey multiple comparison test, \*p<0.05, \*\*p<0.01, \*\*\*p<0.001. (**C–F**) AneuFinder plots revealing perfect euploidy in control thymus (45 freshly isolated T-cells (**C**)) and recurrent chromosomal abnormalities as well as intratumor karyotype heterogeneity in three *Lck-Cre::Mad2l1^{f/f}::Trp53^{f/f}* T-ALLs for which 46 (**D**), 44 (**E**) and 43 (**F**) primary tumor cells were analyzed by single cell sequencing, respectively. Colors refer to chromosome copy number state.
The following figure supplement is available for figure 6:

**Figure supplement 1.** Heterogeneity and aneuploidy scores for control thymus and individual T-ALLs analyzed by single cell sequencing.

**Table 2.** Evolutionary relationship of tumors within the same animal. Focal copy number variants were compared between tumors using the Jaccard Index and tested for significance by transforming the Jaccard Indices into Z-scores (1.96 cutoff for significance). Z-scores were calculated for every tumor individually. Thus, each tumor pair has two Z-scores and was considered weakly related if one comparison was significant (*) and strongly related if both comparisons were significant (**).

| Mouse | Tumor | Z score | Tumor | Z score | Related |
|---|---|---|---|---|---|
| 7122 | HCA1 | 1.08 | HCC1 | 1.36 | |
| 2985 | HCA1 | 1.1 | HCC1 | 1.15 | |
| 6717 | HCA1 | 1.55 | HCC1 | 1.37 | |
| 6705 | HCA1 | 0.13 | HCC1 | 0.56 | |
| 6718 | HCA1 | 1.88 | HCC1 | 0.46 | |
| 6546 | HCA1 | .62 | HCC1 | 1.47 | |
| 6891 | HCA1 | 1.21 | HCC1 | 1.21 | |
| 6891 | HCC1 | 2.81 | HCC2 | 1.37 | * |
| 6755 | HCA1 | 2.49 | HCC1 | 2.69 | ** |
| 6228 | HCA1 | 2.5 | HCC1 | 2.03 | ** |

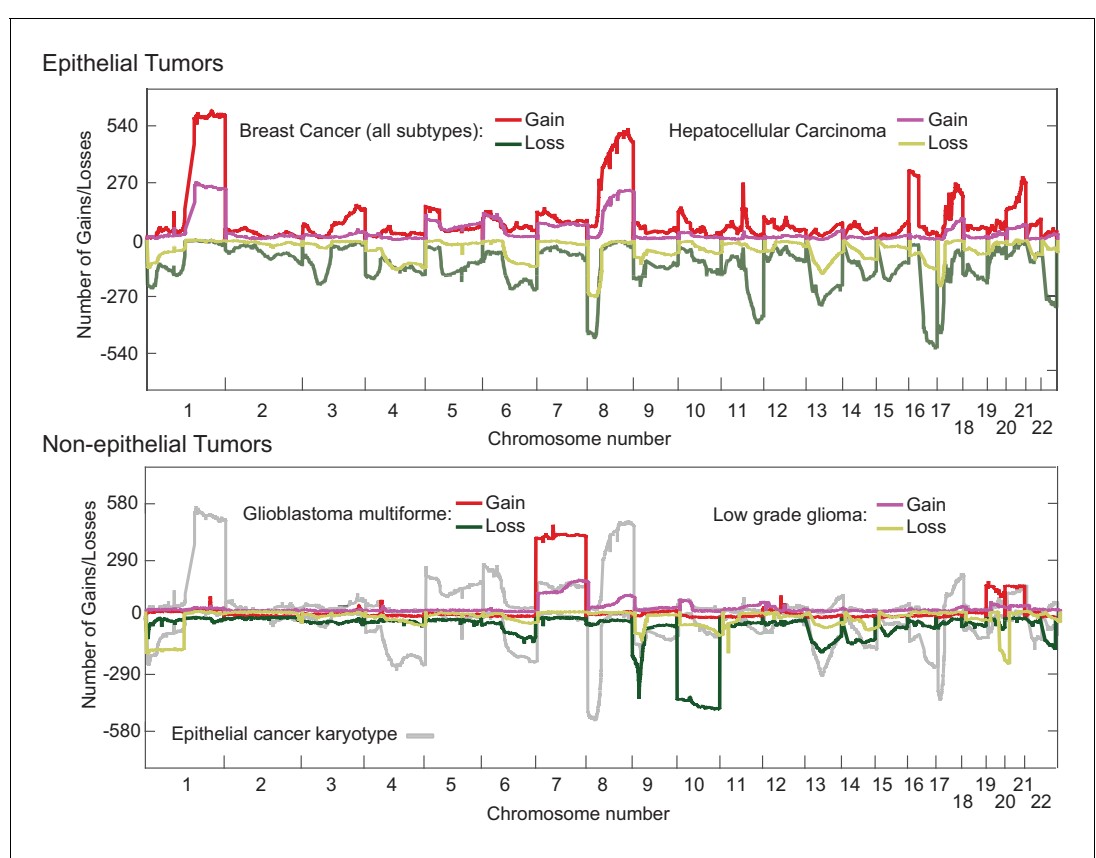

**Figure 7.** Overlaid TCGA copy number data at kilobase resolution for epithelial (Hepatocellular carcinoma and Breast Cancer, upper panel) and non-epithelial (Glioma and Glioblastoma multiforme, lower panel) represented as the number of tumors above a log2 threshold of 0.3 for gains and below −0.3 for losses. The Y-axis is scaled to the total number of tumors analyzed. Analyzed data was sequenced at the Broad institute (Boston, MA) on SNP6.0 chips.

existence of a pan-cancer karyotype. However, non-epithelial cancers exhibit a very different karyotypic pattern than carcinomas (compare upper and lower panels *Figure 7*) and see also (*Zack et al., 2013*). Thus, epithelial and non-epithelial human cancers are similar to Mad2l1-null HCC and T-ALL in having distinct karyotypes, perhaps because they experience different selective pressures.

## Discussion

In this paper we describe the consequences of deleting *Mad2l1*, a core component of the spindle assembly checkpoint, in murine thymocytes and hepatocytes. In both cell types, rapidly growing tumors arise from cells that have lost *Mad2l1* expression as judged from genomic DNA, RT-PCR and Western blotting. Tumorigenesis is promoted in both thymocytes and the liver by heterozygous or homozygous deletion of *Trp53*, reflecting the ability of Trp53 loss to tolerize cells to SAC inactivation as well as the role of Trp53 as a potent tumor suppressor (*Foijer et al., 2014*; *Fujiwara et al., 2005*; *Baker et al., 2009*). Deletion of both *Mad2l1* and *Trp53* in T-cells (promoted by *Lck-Cre*) results in T-cell acute lymphoblastic lymphoma and death of animals within 4–5 months as a result of grossly enlarged thymi and dyspnea. Mice that lose *Trp53* and *Mad2l1* in hepatocytes (promoted by *Alb-Cre*) initially develop regeneration nodules, a sign of on-going liver repair, then hepatocellular carcinoma, a benign liver tumor and finally hepatocellular carcinoma, an aggressive liver cancer resulting in death between 12–15 months of age. HCCs lacking *Mad2l1* grow rapidly for solid tumors, with an estimated doubling time of 28 days. The age of tumor onset and of death in single and double knockout animals strongly suggests that tumorigenesis is driven by the compound effects of *Mad2l1* and *Trp53* loss. However, liver tumors can also arise in mice that are deleted for *Mad2l1* alone, showing that *Trp53* deletion is not a pre-requisite for tumorigenesis. This may relate to the fact that hepatocytes are normally polyploid, and thus, potentially more tolerant of changes in chromosome number than other cell types. We conclude that loss of *Mad2l1* in adult tissues can be strongly oncogenic, in contrast to heterozygous *Mad2l1* deletion, which is only weakly cancer promoting (*Michel et al., 2001*). *Mad2l1^{f/f}* mice having a range of *Trp53* genotypes therefore represent a valuable tool for studying the SAC in vivo.

### Conditional Mad2l1 deletion as a means to study chromosome instability and tumor evolution

The rapid growth of Mad2l1-deficient liquid and solid tumors stands in contrast to previous data showing that Mad2l1 is essential for the survival of cancer cell lines and for development of early mouse embryos (*Kops et al., 2004*; *Dobles et al., 2000*). Mad2l1 loss substantially accelerates tumor development in Trp53-positive and/or mutant backgrounds (depending on cell type) emphasizing that checkpoint inactivation and CIN are not just tolerated by cells, they accelerate disease onset in animals already mutant for a potent tumor suppresser. In cultured MEFs and thymocytes assayed in vivo, the microtubule poisons nocodazole and taxol provoke a significantly weaker cell cycle arrest than in wild-type cells. The near-complete absence of response to nocodazole in MEFs is consistent with checkpoint inactivation but in vivo experiments with thymocytes reveal partial responsiveness to taxol. It is unclear whether this reflects the presence of a Mad2l1-independent pathway for detecting and responding to spindle damage, or the presence of unrecombined Mad2l1-positive cells that have a normal checkpoint response; additional single-cell experiments will be required to resolve this matter.

Parallel experiments in our labs have shown that checkpoint deficiency is also tolerated by basal epidermal skin cells but is lethal to hair follicle stem cells (*Foijer et al., 2013*). These data established that SAC inactivation is compatible with proliferation of some cell types but not others. However, our previous hypothesis that the SAC would be required in highly proliferating cells appears not be true. Since most T-ALLs are clonal and show no evidence of Mad2l1 expression, they must arise from a single checkpoint-null initiating cell following many rounds of cell division. Further experiments with *Mad2l1^{f/f}* animals and Cre drivers should reveal which tissues tolerate checkpoint loss and which do not. The consequences of Mad2l1 inactivation for chromosome structure also appear to differ with cell type: in Mad2l1-deficient HCCs we observe fewer and smaller lesions that in T-ALL. The reasons for this are unknown and the result is subject to the caveat that T-ALLs are clonal and HCC polyclonal. Nonetheless, the data suggest that the consequences of SAC inactivation can vary,

providing a rationale for expanding the study of chromosome segregation in human cells from a few transformed lines to multiple primary cell types.

Loss of Mad2l1 creates a mutator phenotype in the absence of an overt oncogenic driver. In the current study we detect frequent loss of known tumor suppressor genes, *Pten* in T-ALL for example, and amplification of known oncogenes, such as *MET* in HCC. CNVs are also found in many other genes whose function in cancer remains unknown. Large-scale genomic analysis of Mad2l1-deficient tumors may be a generally useful means for identifying new oncogenes and tumor suppressors as well as genes (such as Trp53) whose mutation tolerizes cells to checkpoint loss and aneuploidy (*Torres et al., 2010*; *Fujiwara et al., 2005*). *Mad2l1*<sup>f/f</sup> animals may also be useful in studying the role of genomic instability in drug resistance and tumor recurrence (*Burrell and Swanton, 2014b*, *2014a*) and in the clonal evolution of tumors (*Sotillo et al., 2010*).

## Apparent stabilization of cancer karyotypes in the face of ongoing CIN

Single-cell sequencing of T-ALLs from Mad2l1-deficient mice reveals extensive aneuploidy and intra-tumor karyotypic heterogeneity. In some animals with liver cancer, we found evidence of clonal progression from HCA to HCC as well as genetic progression among physically distinct HCCs in a single animal. This is precisely what we would expect of cells experiencing ongoing CIN. Our data are also consistent with findings from ultra-deep whole genome sequencing of lung cancers demonstrating extensive intra-tumor heterogeneity at the level of point mutations and structural chromosome abnormalities (*de Bruin et al., 2014*; *Gerlinger et al., 2012*). In these studies, different physical regions from the same tumor were repeatedly sequenced, revealing the presence of shared mutations as well as mutations unique to each region, a pattern consistent with genomic instability.

In Mad2l1-deficient murine T-ALLs, HCAs and HCCs, the average karyotype of cell populations is shown by aCGH to involve pan-cancer and tissue-specific patterns of whole chromosome loss and gain. For example, Chr13 is lost and Chr15 gained in both T-ALL and HCC whereas Chr4 and Chr12 are gained in T-ALL and lost in HCC. In humans, recurrent cancer karyotypes have also been observed and these differ between carcinomas, sarcomas and liquid tumors (*Zack et al., 2013*). Single cell and aCGH data are most easily reconciled by postulating that tumor cells experience ongoing CIN but that specific karyotypes have a selective advantage within the environment of a tumor. Karyotypic variation presumably results in differential loss and gain of oncogenes and tumor suppressors as discussed above, but it also causes large-scale changes in gene expression. In general, genes that are located on chromosomes with abnormal ploidy also change in levels of expression, but even when T-ALL and HCC experience the same change in ploidy for a particular chromosome, GSEA shows that differentially expressed genes do not overlap. We therefore speculate that karyotypic selection is imposed both at the level of structural rearrangements in specific genes and broad changes in gene expression. Selection in the face of ongoing *CIN* contrasts with a model of a genome restabilization in which *CIN* is a transient phenomenon and tumor karyotype maintained because aneuploid or structurally abnormal chromosomes are subsequently transmitted with good fidelity. However, further analysis of tumor cells passaged in culture or in syngeneic animals will be required to distinguish unambiguously among these two possibilities.

## Materials and methods

### Generation of conditional knockout mice for *Mad2l1* and *Mad2l1::Trp53* and genotyping

The conditional targeting vector (shown in *Figure 1a* and Extended Data *Figure 1*) was constructed to delete a genomic fragment containing exons 2 and part of exon 5 of the *Mad2l1* gene by homologous recombination. One loxP site was introduced into intron 1 and the other loxP site together with Frt-Neo-Frt cassette was inserted into exon 5, such that *Mad2l1* exon 2 and part of exon 5 were flanked by the loxP sites. Cre-mediated deletion will remove entire exons 2,3,4 and part of exon 5 including stop codon, producing a *Mad2l1*<sup>Δ</sup> allele. Embryonic stem cells derived from 129Sv mice were transfected and selected by genomic southern blot. Homologous recombinant clones were isolated and the loxP-flanked PGKneo cassette was excised by transient expression of FLP recombinase. Chimeric mice were created by injecting Mad2l1-targeted ES cell line (from 129 background)

into C57BL/6 blastocysts generated by superovulation. Chimeras were crossed to C57BL/6 wild-type animals to generate founder lines.

*Mad2l1*[f/f] mice were crossed to *Lck-Cre* or *Alb-Cre* transgenic mice to generate T cell specific and parenchymal liver cell-specific knockouts of Mad2l1 respectively. *Lck-Cre* and *Alb-Cre::Mad2l1*[f/f] mice were then inter-crossed with *Trp53*[f/f] mice (*Jonkers et al., 2001*). All animals were kept in pathogen-free housing under guidelines approved by the Center for Animal Resources and Comparative Medicine at Harvard Medical School or at the Wellcome Trust Sanger Institute. Animal protocols were approved by the Massachusetts Institute of Technology, Harvard Medical School Committees on Animal Care (IACUC numbers I04272 and IS00000178), UK Home Office, and UMCG animal facility (DEC 6369).

Tail DNA was isolated using NucleoSpin Tissue Kit from Macherey-Nagel according to the manufacturer's protocol. The following primers were used for *Mad2l1*[+] ASOL233, GCAGACCAAAC-GAACCTAAGTT. ASOL 238, GCAAGAGGTGGTTCAATAGTGAG, *Mad2l1*[f] MOL232, AGGC TGAGCCGGGCCTTAGGAC; MOL233, GTAACCGTGTAATAACGTTTAAGTCTC, *Mad2l1*[Δ] MOL231, GTCTGCGGTGAGGTTGG; ASOL233, GCAGACCAAACGAACCTAAGTT. *Alb-Cre* AlbF, GTTAA TGATCTACAGTTATTGG and AlbR, CGCATAACCAGTGAAACAGCATTGC. *Lck-Cre* LckF, CCTTGG TGGAGGAGGGTGGAATGAA, LckR, TAGAGCCCTGTTCTGGAAGTTACAA, and CreT2R, CGCA TAACCAGTGAAACAGCATTGC.

## Cell culture

MEFs were isolated as described previously (*Foijer et al., 2005*) and cultured in DMEM containing 10% FCS, pyruvate, non-essential amino acids and penicillin/ streptomycin (Invitrogen). Cells were genotyped using the above-mentioned primers to confirm genotypes and routinely tested for mycoplasma contamination. For spindle checkpoint integrity measurements, cells were exposed to 250 ng/ml nocodazole (Sigma) for 4–6 hr, fixed in 70% ethanol and labeled with Alexa Fluor-488-conjugated pHistoneH3 antibodies (Cell Signaling, RRID:AB_10694488) as described previously (*Foijer et al., 2014*). For time-lapse imaging, cells were transduced with H2B-GFP (*Foijer et al., 2014*) as described previously (*Foijer et al., 2005*) and seeded on 4-well imaging slides (LabTek, Thermo Fisher) in the presence of 250 ng/ml nocodazole (Sigma). Cells were imaged on a DeltaVision Elite imaging station (GE Healthcare).

## Histology

Animals were euthanized and their thymus or liver were removed and rinsed in PBS. Tissues collected were fixed overnight in formalin. Fixed tissues were then stored in 70% ethanol until they were embedded in paraffin. Section slides were prepared and standard H&E staining were done at Rodent Histopathology Core facility at Dana-Farber/Harvard Cancer Center.

## Western blots and antibodies

Protein from tumors was isolated using protein lysis buffer (Millipore) in the presence of protease inhibitors (Millipore). Protein concentration was quantified using the Bradford assay (Biorad). 20 μg of total protein was run on a 4–12% gradient gel (Invitrogen) per sample and blotted on PVDF membrane (Millipore). Antibodies used were mouse monoclonal Mad2l1 (BD Biosciences, RRID:AB_ 398005), mouse monoclonal Actin (Cell Signaling, RRID: AB_2223172) and HRP-labeled goat-anti mouse (New England Biolabs).

## Array-based comparative genomic hybridization and single cell sequencing

Mouse thymus and liver genomic DNA was extracted with NucleoSpin Tissue Kit (Macherey-Nagel). Sex-mismatched wild type liver DNA was used as control. Mouse Genome CGH Microarrays 44K or 244K from Agilent were used. Array hybridization and data analysis were performed at the Wellcome Trust Sanger Institute, the Partners HealthCare Center for Personalized Genetic Medicine at Harvard Medical School, and the BioMicro Center at the Massachusetts Institute of Technology. Low coverage Next-Gen sequencing of liver tumor DNA isolated as described above was performed at the BioMicro Center at the Massachusetts Institute of Technology.

For single cell sequencing, T-ALL samples or primary thymus were dissected and homogenized through a tissue strainer. Single cells in G1 were sorted into 96 wells plate by flow cytometry using a Hoechst/Propidium iodide double staining. Cells were then lysed, DNA sheared and DNA was barcode labeled followed by library preparation as described previously (*van den Bos et al., 2016*) in an automated fashion (Agilent Bravo robot). Single cell libraries were pooled an analyzed on an Illumina Hiseq2500 sequencer. Single cell sequencing data was analyzed using AneuFinder software as described previously (*Bakker et al., 2016*).

### RT-PCR, qPCR, and expression arrays

RNA was isolated using the RNeasy kit (Qiagen). For qPCR reactions, 1 µg of total RNA was used for a reverse transcriptase reaction (Superscript II, Invitrogen). The resulting cDNA was used as a template for qPCR (ABI PRISM 7700 Sequence Detector) in the presence of SYBR-green (Invitrogen) to label the product. The relative amounts of cDNA were compared to Actin to correct for the amount of total cDNA. Average values and standard deviations were calculated as indicated in Figure legends and compared to the expression values in control mice (normalized to the value of 1). We used the following primers:

Trp53 A Fw TGTTATGTGCACGTACTCTCC, Trp53 A Rv GTCATGTGCTGTGACTTCTTG
Trp53 B Fw TCCGAAGACTGGATGACTG, Trp53 B Rv AGATCGTCCATGCAGTGAG,
Mad2l1 A Fw AAACTGGTGGTGGTCATCTC, Mad2l1 A Rv TTCTCTACGAACACCTTCCTC,
Actin A Fw CTAGGCACCAGGGTGTGATG, and Actin A Rv GGCCTCGTCACCCACATAG.
Illumina expression microarrays were performed as described previously (*Foijer et al., 2014*).

### Public availability of high throughput data

Array data is publically available via GEO accession numbers GSE63689 and GSE63100. Sequencing data of liver tumors is available at the Sequence Read Archive accession number SRA191233. Single cell sequencing data of lymphomas is available at the Sequence Read Archive accession numbers under GSE63689.

### Plots and statistical analysis

Graphing plots and statistical testing was performed using GraphPad Prism (GraphPad Software) or MATLab (Mathworks).

## Acknowledgements

We thank the members of the Sorger, Bradley and Foijer labs for fruitful discussion. MRI was performed at the Longwood Small Animal Imaging Facility (LSAIF), Beth Israel Hospital. Elaine Lunsford (LSAIF) optimized the EOVIST protocol. This work was funded by NIH grants CA084179 and CA139980 to PKS, a research contract from Vertex Pharmaceuticals to PKS, Wellcome Trust funding to AB, European Research Council Advanced grant (ROOTS-Grant Agreement 294740) to PML, and EMBO, Dutch Cancer Society (RUG-2012–5549) and Stichting Kinder Oncologie Groningen (SKOG) funding to FF.

## Additional information

### Funding

| Funder | Grant reference number | Author |
|---|---|---|
| National Institute for Health Research | CA084179 | Lee A Albacker<br>Ying Yue<br>Stephanie Davis<br>Peter K Sorger |
| National Institute for Health Research | CA139980 | Lee A Albacker<br>Ying Yue<br>Stephanie Davis<br>Peter K Sorger |
| KWF Kankerbestrijding | 2012-RUG-5549 | Floris Foijer |

| | | Bjorn Bakker |
|---|---|---|
| H2020 European Research Council | ERC advanced ROOTS | Diana C Spierings Peter M Lansdorp |
| European Molecular Biology Organization | Longterm fellowship | Floris Foijer |
| Stichting Kinder Oncologie Groningen | | Floris Foijer |

The funders had no role in study design, data collection and interpretation, or the decision to submit the work for publication.

### Author contributions

FF, Conceptualization, Supervision, Funding acquisition, Investigation, Writing—original draft, Writing—review and editing; LAA, Conceptualization, Investigation, Visualization, Methodology, Writing—original draft; BB, Data curation, Formal analysis, Investigation; DCS, Methodology, Single cell sequencing; YY, Investigation, Methodology, Data analysis; SZX, Resources, Initial characterisation Mad2 conditional knockout; SD, Investigation, Methodology; AL-J, Resources, Engineering conditional Mad2l1 knockout allele; DT, BH, Resources, Methodology; BF, Resources, Investigation; RTB, Formal analysis, Mouse pathology; PML, Funding acquisition, Provided access to single cell sequencing; AB, Supervision, Funding acquisition; PKS, Conceptualization, Supervision, Funding acquisition, Methodology, Writing—original draft, Writing—review and editing

### Author ORCIDs

Floris Foijer, iD http://orcid.org/0000-0003-0989-3127
Stephanie Davis, iD http://orcid.org/0000-0002-0022-4210

### Ethics

Animal experimentation: All animals were kept in pathogen-free housing under guidelines approved by the Center for Animal Resources and Comparative Medicine at Harvard Medical School or at the Wellcome Trust Sanger Institute. Animal protocols were approved by the Massachusetts Institute of Technology, Harvard Medical School Committees on Animal Care (IACUC numbers I04272 and IS00000178), UK Home Office, and UMCG animal facility (DEC 6369).

## Additional files

### Major datasets

The following datasets were generated:

| Author(s) | Year | Dataset title | Dataset URL | Database, license, and accessibility information |
|---|---|---|---|---|
| Foijer F | 2017 | Copy number changes (average and single cell) and matching transcriptomes of HCCs and T-ALLs isolated from Mad2 p53 conditional double knockout mice | https://www.ncbi.nlm.nih.gov/geo/query/acc.cgi?&acc=GSE63689 | Publicly available at the NCBI Gene Expression Omnibus (accession no: GSE63689) |
| Albacker LA | 2015 | Cytogenetic aberrations in Hepatocellular adenoma and carcinoma | https://www.ncbi.nlm.nih.gov/geo/query/acc.cgi?acc=GSE63100 | Publicly available at the NCBI Gene Expression Omnibus (accession no: GSE63100) |
| Albacker LA | 2015 | Hepatocellular adenoma/carcinoma from Mad2 deficient hepatocytes | https://www.ncbi.nlm.nih.gov/sra/?term=SRA191233 | Publicly available at the NCBI Sequence Read Archive (accession no: SRA191233) |

The following previously published dataset was used:

| Author(s) | Year | Dataset title | Dataset URL | Database, license, and accessibility information |
|---|---|---|---|---|
| National Cancer Institute | 2017 | TGCA | https://portal.gdc.cancer.gov/search/s?facetTab=cases | Publicly available at GDC Data Portal |

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
