## [Decision Letter]

[Editors’ note: this article was originally rejected after discussions between the reviewers, but the authors were invited to resubmit after an appeal against the decision.]

Your manuscript entitled "Maintenance of a stable karyotype in Mad2-null lymphoma and liver cancer cells by stabilizing selection" has now been seen by three reviewers. Their comments are appended below. As you can see all three reviewers thought that the experiments were carefully executed and the data are compelling. They also stated that the analysis of the spindle checkpoint in different tissues is very interesting and the cancer phenotypes/karyotypes generated by the same perturbation are revealing. However, while one reviewer was supportive of publication, the other two felt that the paper lacked novelty, in that it does not go much beyond previous work from your lab. They also felt that some of the conclusions were overstated. Specifically, their concerns were:

Lack of novelty:

1) The manuscript follows similar work by your lab in skin and, using a different checkpoint mutant (an engineered allele of the kinase Mps1), in T-cells (2 prior PNAS papers). These prior PNAS papers established non-essentiality of the checkpoint in a specific tissue context and described a cancerous phenotype (lymphoma) associated with reducing spindle checkpoint activity in T-cells.

2) The central findings – that CIN + loss of p53 results in tumorigenesis and that cancers display tissue-specific patterns of chromosomal alterations have been reported previously. p53 loss has been found to accelerate cancer development in Mad1 and Mad2-mutant mice (Chi et al., IJC 2009), in Cdc20 and BubR1-mutant mice (Li et al., PNAS 2010), in the authors' own previous work with Mps1-mutant mice (PNAS 2014), and in several other mouse models. Tissue-specific patterns of chromosome gains and losses have also been described (e.g., Ozery-Flato et al., Genome Biology 2011; Duijf et al., IJC 2012; Zach et al., Nature Genetics 2013; and others).

Overstatement of conclusions:

1) The claim made in the title on 'stabilizing selection' is not supported by the data. The authors demonstrate that single T-ALL cells display varying karyotypes, while a bulk tumor population shows evidence of clonal abnormalities. From this, they infer that stabilizing selection is taking place. The authors' evidence for stabilizing selection in HCC's is even weaker, as the karyotypes of the HCCs are generally not significantly correlated with the karyotypes of the HCA precursor lesions.

2) The statement that "T-cells and hepatocytes survive checkpoint loss, an event hitherto assumed to be lethal." is not accurate. The authors have previously shown that epidermal cells tolerate loss of Mad2 (Foijer et al., PNAS 2013).

In light of these reservations, I, regretfully, have to reject the paper. It is my hope that you find the reviewers' comments useful for submission of the manuscript elsewhere.

*Reviewer #1:*

This manuscript addresses the role of the spindle assembly checkpoint protein Mad2 in tumorigenesis. Foijer et al. examine the consequences of inactivating MAD2 and p53 in two tissues, T cells and the liver. The authors find that inactivation of both MAD2 and p53 leads to rapid onset of ALL in T cells and hepatocellular carcinoma. Single cell sequencing of these tumors shows that cells lacking MAD2 and p53 are as expected genomically unstable but sequencing of whole tumors reveals a recurrent, perhaps even clonal karyotype indicating that specific tumorigenic karyotypes are selected for.

This is a nice paper. The data are compelling and support the conclusion. A key question that remains is whether aneuploidy (whole chromosome gains) or specific subchromosomal alterations (CNVs and point mutations) drive the disease. For example, the authors show very nicely that PTEN loss occurs and that the MET locus is overexpressed. Isn't it much more likely that these specific changes in the activity of known tumor suppressor genes and oncogenes are what is driving tumorigenesis, rather than a 50 percent change in gene expression (and even less in the polyploid liver) that occurs through the gain or loss of an entire chromosome? The paper could benefit from a discussion of what exactly drives tumorigenesis in cells with CIN.

The following scientific points should also be addressed:

1) What is the cause of death in Mad2-/- p53+/- animals?

2) The observation that Mad2 inactivation on its own causes HCC is interesting. It suggests that in a polyploid tissue, CIN is more tumorigenic. This should be discussed.

3) The observation that some MAD2 loci are intact in HCC is peculiar. The authors propose that the functional MAD2 could be present in cells other than hepatocytes. The authors could easily test this hypothesis by sorting hepatocytes prior to the analysis of the MAD2 locus.

4) Aneuploidies here are shown as box plots. This can be misleading. Because, whole chromosome aneuploidy can lead to secondary genome instability events, it is important to show that the changes in gene copy number indeed affect the whole chromosome and that the increases are not driven by amplifications/deletions of parts of the chromosomes. The authors should show the karyotypes of the tumors in the supplement.

5) The finding that the same oncogene cocktail (p53 loss and Mad2 loss) causes different tumor karyotypes in different tissues is important. It indicates that there is no pan-cancer karyotype driven by oncogene gains and losses but rather that tissue of origin is a key determinant of a cancer karyotype. Indeed, when querying all cancer types in the TCGA data set, one sees highly characteristic recurrent patterns of chromosome gains and losses in specific tumor types that have little resemblance to the pan cancer karyotype. This point should be strengthened and flushed out.

6) If loss of MAD2 is so oncogenic, why is it not observed in tumors? Most tumors have lost p53 pathway function so MAD2 inactivation should be a frequent occurrence.

7) The purifying selection hypothesis is interesting and certainly consistent with the observation that aneuploid karyotypes are selected against in regenerating tissues of BubR1H/H mutant animals in vivo. The authors may want to cite this paper (Pfau et al., 2016) to strengthen their argument.

*Reviewer #2:*

In this manuscript, Foijer et. al generate deletions of the gene encoding the crucial SAC component Mad2 in the liver and T-cell lineages. The resulting mice develop hepatocellular carcinoma and T-ALL, respectively, and disease progression is accelerated in a p53-null genetic background. They report that the HCC's and ALL's that develop in these animals harbor distinct karyotypes, in spite of ongoing CIN caused by the MAD2 deletion. In general, the experiments presented in this paper are well-executed and support many of the conclusions drawn by the authors. However, I do not believe that this paper provides either enough novelty or mechanistic insight to warrant publication in *eLife*. My main concerns are as follows:

1) The central findings in this paper – that CIN + loss of p53 results in tumorigenesis and that cancers display tissue-specific patterns of chromosomal alterations – are not particularly surprising. P53 loss has been found to accelerate cancer development in Mad1 and Mad2-mutant mice (Chi et al., IJC 2009), in Cdc20 and BubR1-mutant mice (Li et al., PNAS 2010), in the authors' own previous work with Mps1-mutant mice (PNAS 2014), and in several other mouse models. Tissue-specific patterns of chromosome gains and losses have also been thoroughly described (e.g., Ozery-Flato et al., Genome Biology 2011; Duijf et al., IJC 2012; Zach et al., Nature Genetics 2013; and many others).

2) In some cases, I believe that the authors misrepresent what is known about aneuploidy and cancer in order to over-emphasize the novelty of their work. For instance, in the first paragraph of the Introduction, they write "For different tumor types, the pattern of chromosome gain and loss appears remarkably constant (Davoli et al., 2013)." This is not reported in the paper that they cite, and is false. In the last paragraph of the Introduction, they write "T-cells and hepatocytes survive checkpoint loss, an event hitherto assumed to be lethal." Yet, the authors have previously shown that epidermal cells tolerate loss of Mad2 (Foijer et al., PNAS 2013), and a variety of experiments with other models of CIN (Mad2 overexpression, Mps1 mutation, Cdc20 mutation) also have shown that on-going CIN can be tolerated.

3) The authors demonstrate that single T-ALL cells display varying karyotypes, while a bulk tumor population shows evidence of clonal abnormalities. From this, they infer that "stabilizing selection" is taking place. The authors' evidence for stabilizing selection in HCC's is even weaker, as the karyotypes of the HCCs are generally not significantly correlated with the karyotypes of the HCA precursor lesions. If the authors wished to strengthen their case for "stabilizing selection", then they could derive cultures from the single T-ALL cells, karyotype them over time, and demonstrate that the clones in fact evolve towards the karyotype displayed by the bulk tumor. A direct observation of stabilizing selection in this manner I believe would be quite interesting, but as it stands the title of the paper is not well-supported by the data.

Reviewer #3:

This study reports analysis of conditional deletion of the spindle assembly checkpoint component Mad2 in two different tissue contexts – T-cells and liver cells. In both cases, analysis with and without concomitant p53 deletion is performed. The manuscript reports a large body of in vivo mouse work – the most interesting findings relate to the tumor phenotypes and the spectrum of aneuploidies observed in the different contexts. This manuscript follows similar work conducted by the same group of authors in skin and, using a different checkpoint mutant (an engineered allele of the kinase Mps1), in T-cells (2 prior PNAS papers). Overall, the work quality is high (although I cannot technically judge the informatics analysis of the CGH data that is presented in the adenoma-carcinoma comparison in liver and some of the data-conclusion relationships would benefit from further exposition/clarification for a non-expert audience). While the findings described here are undoubtedly valuable to the ongoing analysis of aneuploidy induction and cancer phenotypes, the manuscript is descriptive, the prior PNAS papers established non-essentiality of the checkpoint in a specific tissue context and a lymphoma phenotype associated with reducing spindle checkpoint activity in T-cells, and there is little attempt to analyze why specific tissues tolerate Mad2 loss (in their first such paper in skin, they observed follicle stem cells did not tolerate Mad2 loss but epidermal cells did – why is unclear). In addition, it is not clear to me where the claim made in the title on 'stabilizing selection' arises from – in contrast, the difference in aneuploidy patterns in the two contexts seems clearer. With T-cells, which can be induced to proliferate, it seems feasible to analyze the effect of checkpoint inhibition in vitro (the authors state "…adult T cells from these animals developed normally and proliferated upon stimulation in vitro, showing that T cells are tolerant of Mad2 loss." but do not show any data associated with this statement). Do these cells have a naturally different paced mitosis that allows enough time to get things right? Or are they able to eliminate errors efficiently? Or do they efficiently eliminate cells that underwent error-prone division?) This is not to diminish the value of the work done but, if resources are not an issue, one could keep using Cre drivers in different tissues, delete Mad2, and analyze the effects and it seems that some additional effort to understand why outcomes are different is important to make the work appeal broadly. Here, this seems most straightforward to do with T-cells, which can be stimulated to proliferate in vitro.

---

## [Author Response]

[Editors’ note: the author responses to the first round of peer review follow.]

*[…] Reviewer #1:*

*[…] This is a nice paper. The data are compelling and support the conclusion. A key question that remains is whether aneuploidy (whole chromosome gains) or specific subchromosomal alterations (CNVs and point mutations) drive the disease. For example, the authors show very nicely that PTEN loss occurs and that the MET locus is overexpressed. Isn't it much more likely that these specific changes in the activity of known tumor suppressor genes and oncogenes are what is driving tumorigenesis, rather than a 50 percent change in gene expression (and even less in the polyploid liver) that occurs through the gain or loss of an entire chromosome? The paper could benefit from a discussion of what exactly drives tumorigenesis in cells with CIN.*

This is an excellent point – we agree with the reviewer that the determining the relative roles of CNVs and whole-chromosome amplification in tumorigenesis (e.g. Williams et al., Science 2008; Foijer et al., PNAS 2014) remains a challenge. We did not mean to imply that CNVs and point mutations were not responsible for tumorigenesis although it does seem possible that changes in chromosome number (and the attendant modest change in RNA abundance due to ploidy changes) also play a role. We have added a section to the Discussion describing how CIN-induced DNA damage (for instance as shown by Janssen et al., 2011, Science) can lead to structural alterations in cancer cells genomes. This could in principal explain how CIN results in PTEN loss MET amplification.

*The following scientific points should also be addressed:*

*1) What is the cause of death in Mad2-/- p53+/- animals?*

Most of these animals, particularly those that die up to age 12 mo. die of T-ALL; we now make this clear in the body of the text.

*2) The observation that Mad2 inactivation on its own causes HCC is interesting. It suggests that in a polyploid tissue, CIN is more tumorigenic. This should be discussed.*

We now briefly discuss this possibility in light of the fact that hepatocytes have an intrinsic tendency to become polyploid. As the reviewer suggests, they probably tolerate aneuploidy better than other cell lineages and thus may not require further predisposing mutations to proliferate in a CIN background. As the reviewer will appreciate however, this remains a speculation and it is therefore difficult to do more than point out the possible implications.

*3) The observation that some MAD2 loci are intact in HCC is peculiar. The authors propose that the functional MAD2 could be present in cells other than hepatocytes. The authors could easily test this hypothesis by sorting hepatocytes prior to the analysis of the MAD2 locus.*

In our opinion, finding non-recombined Mad2 loci in complex, solid tumors that invade adjacent normal tissue is not at all unexpected. Our current interpretation is either (i) that wild-type MAD2 loci are present in the tumor sample due to the presence of other liver cell types or infiltrating immune cells or (ii) they arise from hepatocytes present in adjacent tissue in which recombination is incomplete; this is expected since HCCs are highly invasive and cannot be cleanly dissected away from normal tissue.

However, we suspect that the question the reviewer is actually asking is whether cells that are heterozygous for MAD2 deletion can also be tumor cells. We think that this is unlikely because germline Mad2 heterozygosity does not cause HCC either alone or in combination with p53 deletion (Figure 2 and Figure 2—figure supplement 1). Actually isolating these cells by FACs, as suggested by the reviewer, is not straightforward since HCCs are highly fibrotic and isolation would need to be followed by a determination of transformed status. We do not have methods in place to do this nor are we able to develop the methods in the near future because HCC-bearing animals are not currently available; generating more will take up to a year due to slow tumor progression. We do not believe that waiting for this result is warranted, since it will not substantially change our overall conclusions.

*4) Aneuploidies here are shown as box plots. This can be misleading. Because, whole chromosome aneuploidy can lead to secondary genome instability events, it is important to show that the changes in gene copy number indeed affect the whole chromosome and that the increases are not driven by amplifications/deletions of parts of the chromosomes. The authors should show the karyotypes of the tumors in the supplement.*

In response we have now added a large number of aCGH plots for HCCs and T-ALLS in Figure 5, and Figure 5—figure supplement 1 and B that show that we primarily observe whole chromosome abnormalities in our animals. We have also plotted our single cell sequencing data using the recently developed algorithm AneuFinder (Bakker et al., Genome Biology, 2016) (Figure 6), which further demonstrates the high prevalence of whole-chromosome events and addresses the reviewer’s concern.

*5) The finding that the same oncogene cocktail (p53 loss and Mad2 loss) causes different tumor karyotypes in different tissues is important. It indicates that there is no pan-cancer karyotype driven by oncogene gains and losses but rather that tissue of origin is a key determinant of a cancer karyotype. Indeed, when querying all cancer types in the TCGA data set, one sees highly characteristic recurrent patterns of chromosome gains and losses in specific tumor types that have little resemblance to the pan cancer karyotype. This point should be strengthened and flushed out.*

This is an excellent point that was dealt with more explicitly in an earlier draft of our paper (and has now been brought back in the current revision). In discussions with colleagues, we found that the entire concept of a pan-cancer karyotype was quite controversial for precisely the reason’s the reviewer describes. In our revised manuscript, we are more explicit in saying that Mad2 loss drives tissue specific karyotypic changes, which argues against a pan-cancer karyotype, and that TGCA data (Figure 7) are also consistent with tissue selectivity. In the Discussion section we have expanded the text on tissue specificity, explaining that different cell types benefit from different genetic contexts, and have correctly cited the Davoli et al. paper (acknowledging that they too observed tissue specific effects on chromosome ploidy).

*6) If loss of MAD2 is so oncogenic, why is it not observed in tumors? Most tumors have lost p53 pathway function so MAD2 inactivation should be a frequent occurrence.*

This is a good point: Mad2 mutations (and mutations in most other spindle assembly checkpoint genes) are exceedingly rare in human cancer. However, if you consider signaling pathways more generally, it is often very difficult to predict which genes will be mutated oncogenes (or tumor suppressors) and which will not. Mad2 is an E2F target gene, and thus is frequently overexpressed in cancers carrying mutations in the retinoblastoma (Rb) pathway (about 60% of all cancers). Targeted over-expression of Mad2 results in ongoing CIN, so it is possible that this is the more common means of inducing aneuploidy in humans. Indeed, the ability of Rb loss to cause CIN has been interpreted as arising from Mad2 overexpression (Hernando et al., Nature 2004) and consequent cohesion defects (van Harn et al., Genes and Dev, 2010). With respect to the current work it remains true that targeted inactivation of Mad2 is a selective and powerful way to provoke ongoing CIN in vivo and to study the consequences, the main purpose of developing the Mad2 mouse model. We have revised the Discussion to address these issues.

*7) The purifying selection hypothesis is interesting and certainly consistent with the observation that aneuploid karyotypes are selected against in regenerating tissues of BubR1H/H mutant animals* in vivo*. The authors may want to cite this paper (Pfau et al., 2016) to strengthen their argument.*

This is a very interesting new paper (arguing that aneuploidy is selected against in a non-transformed setting) that we now cite in the Discussion of single cell sequencing data.

*Reviewer #2:*

*[…] 1) The central findings in this paper – that CIN + loss of p53 results in tumorigenesis and that cancers display tissue-specific patterns of chromosomal alterations – are not particularly surprising. P53 loss has been found to accelerate cancer development in Mad1 and Mad2-mutant mice (Chi et al., IJC 2009), in Cdc20 and BubR1-mutant mice (Li et al., PNAS 2010), in the authors' own previous work with Mps1-mutant mice (PNAS 2014), and in several other mouse models. Tissue-specific patterns of chromosome gains and losses have also been thoroughly described (e.g., Ozery-Flato et al., Genome Biology 2011; Duijf et al., IJC 2012; Zach et al., Nature Genetics 2013; and many others).*

We appreciate and understand this critique and have thoroughly revised the manuscript in response. We clearly did not succeed in communicating the novel aspects of our findings relative to previous work by our own lab and by others. Briefly, the new findings in the current work include the following: (i) complete loss of the spindle assembly checkpoint can be tolerated in tumors in vivo; hitherto, only partial loss of function mutations had been analyzed and tumor latencies were long (ii) SAC-deficient tumors evolve highly recurrent karyotypes despite frequent missegregation events, which we deduce from high grade intratumor heterogeneity as measured in T-ALLs. We have analyzed one additional tumor to substantiate this statement and discuss our findings in comparison to our Mps1 truncation model (Foijer et al., 2014; Bakker et al. 2016). We also calculate heterogeneity scores for all four samples (1 healthy thymus and 3 T-ALLs) and find that these are substantially higher than in T-ALLs driven by an Mps1 truncation allele (Bakker et al., 2016). In the latter study, we found that 1 out of 2 of the primary tumor cells exhibited abnormalities in mitosis, as assessed by time lapse microscopy, further showing that the observed intratumor heterogeneity is indeed a consequence of ongoing CIN; and (iii) resulting karyotypes are tissue-specific. While the latter point is not new (which we now more thoroughly explain and reference in the revised manuscript), our study is the first in which tumor karyotypes arising from the same combination of drivers but in different tissues can be compared directly.

*2) In some cases, I believe that the authors misrepresent what is known about aneuploidy and cancer in order to over-emphasize the novelty of their work. For instance, in the first paragraph of the Introduction, they write "For different tumor types, the pattern of chromosome gain and loss appears remarkably constant (Davoli et al., 2013)." This is not reported in the paper that they cite, and is false. In the last paragraph of the Introduction, they write "T-cells and hepatocytes survive checkpoint loss, an event hitherto assumed to be lethal." Yet, the authors have previously shown that epidermal cells tolerate loss of Mad2 (Foijer et al., PNAS 2013), and a variety of experiments with other models of CIN (Mad2 overexpression, Mps1 mutation, Cdc20 mutation) also have shown that on-going CIN can be tolerated.*

We agree that we over-simplified the findings of the Davoli paper, which indeed states that there is a tissue specific component to the pan-cancer karyotype. We have fixed this error, which arose in editing the paper for length. Our previous work on epidermal cells only showed that relatively differentiated lineages could survive checkpoint loss: in stem cells it was lethal. Finding that the checkpoint is dispensable in a rapidly growing tumor is qualitatively different in our mind. Nonetheless, we agree with the reviewer that we are probably fighting an old battle here and that we have not adequately accounted for the literature from our own lab and others. In part this arises because we had hoped to publish the current paper in 2013, but have had great difficulty convincing reviewers that we had in fact knocked out Mad2! We have made multiple revisions to the text to address this point.

*3) The authors demonstrate that single T-ALL cells display varying karyotypes, while a bulk tumor population shows evidence of clonal abnormalities. From this, they infer that "stabilizing selection" is taking place. The authors' evidence for stabilizing selection in HCC's is even weaker, as the karyotypes of the HCCs are generally not significantly correlated with the karyotypes of the HCA precursor lesions. If the authors wished to strengthen their case for "stabilizing selection", then they could derive cultures from the single T-ALL cells, karyotype them over time, and demonstrate that the clones in fact evolve towards the karyotype displayed by the bulk tumor. A direct observation of stabilizing selection in this manner I believe would be quite interesting, but as it stands the title of the paper is not well-supported by the data.*

We appreciate that our use of the term ‘stabilizing selection’ may not be correct and we have largely eliminated it from the manuscript as part of an overall re-write. We believe that the recurrent karyotypes observed in T-ALLs combined with a high level of intratumour heterogeneity at a single cell level argues in favor of stabilizing selection but agree that the experiment suggested by the reviewer would be more convincing. We have tried multiple times to grow viable Mad2 deficient T-ALL tumor cells in culture, a process that it is not straightforward (establishing even aggressive tumors in long-term culture is hit and miss, particularly for mouse cells that undergo crises after only a few passages). Thus far we have failed to get Mad2 deficient T-ALLs to expand from single cells. By putting less emphasis on this aspect of our findings in the revised manuscript, and by removing the claim from the title, we hope we have struck the right balance in describing the key features of the current work.

Reviewer #3:

*This study reports analysis of conditional deletion of the spindle assembly checkpoint component Mad2 in two different tissue contexts – T-cells and liver cells. In both cases, analysis with and without concomitant p53 deletion is performed. The manuscript reports a large body of* in vivo *mouse work – the most interesting findings relate to the tumor phenotypes and the spectrum of aneuploidies observed in the different contexts. This manuscript follows similar work conducted by the same group of authors in skin and, using a different checkpoint mutant (an engineered allele of the kinase Mps1), in T-cells (2 prior PNAS papers). Overall, the work quality is high (although I cannot technically judge the informatics analysis of the CGH data that is presented in the adenoma-carcinoma comparison in liver and some of the data-conclusion relationships would benefit from further exposition/clarification for a non-expert audience). While the findings described here are undoubtedly valuable to the ongoing analysis of aneuploidy induction and cancer phenotypes, the manuscript is descriptive, the prior PNAS papers established non-essentiality of the checkpoint in a specific tissue context and a lymphoma phenotype associated with reducing spindle checkpoint activity in T-cells, and there is little attempt to analyze why specific tissues tolerate Mad2 loss (in their first such paper in skin, they observed follicle stem cells did not tolerate Mad2 loss but epidermal cells did – why is unclear). In addition, it is not clear to me where the claim made in the title on 'stabilizing selection' arises from – in contrast, the difference in aneuploidy patterns in the two contexts seems clearer. With T-cells, which can be induced to proliferate, it seems feasible to analyze the effect of checkpoint inhibition* in vitro *(the authors state "…adult T cells from these animals developed normally and proliferated upon stimulation* in vitro*, showing that T cells are tolerant of Mad2 loss." but do not show any data associated with this statement). Do these cells have a naturally different paced mitosis that allows enough time to get things right? Or are they able to eliminate errors efficiently? Or do they efficiently eliminate cells that underwent error-prone division?) This is not to diminish the value of the work done but, if resources are not an issue, one could keep using Cre drivers in different tissues, delete Mad2, and analyze the effects and it seems that some additional effort to understand why outcomes are different is important to make the work appeal broadly. Here, this seems most straightforward to do with T-cells, which can be stimulated to proliferate* in vitro.

We thank this reviewer for these comments and have made numerous changes to the text to address them (as described below). Specifically:

We acknowledge that we failed to describe the novelties of our findings and the nature of the advance of previous findings including our own work;

We do not in fact know why some tissue tolerate Mad2 loss and others do not. Based on our previous work on the skin we had assumed that this had to do with how proliferative the cell type was. Our data no longer support this simple explanation, something we now discuss in the manuscript;

We have eliminated the term “purifying selection” from the title of the paper and have extensively revised the discussion of that topic;

With respect to the question of growing Mad2 cells in vitro, this has in fact require a fair amount effort to identify suitable conditions. For several cell types, low oxygen tension appears to be critical, but we have not succeeded as-yet with HCCs. For T-cells we have established that we can grow primary T-cells and Mad2-null T-ALL cells following stimulation, but data with non-transformed Mad2-null primary T-cells are variable. We are as-yet unable to perform the in-depth comparative cell-level studies suggested by the reviewer (and are also removing the statement about Mad2-null, non-transformed T-cells until repeatability improves). This is an area in which we are actively working since it would be very interesting to have Mad2-wt and null cells from different lineages for detailed microscopic analysis. It should also be noted that T-cells are a very challenging cell type in which to perform a detailed analysis of chromosome segregation (relative to fibroblasts) because they are small, round and weakly adherent to microscope slides;

With respect to the comment about resources and additional tissue types, we do not in fact have any grants available to perform studies with additional Cre drivers (although we are hopeful that possible acceptance of this manuscript will help to rectify that);

We find all of the additional questions raised by the reviewer extremely interesting (Do these cells have a naturally different paced mitosis that allows enough time to get things right? Or are they able to eliminate errors efficiently? Or do they efficiently eliminate cells that underwent error-prone division?) and these are precisely the questions we are attempting to address with our on-going research. However, getting definitive answers is likely to take very substantial additional work over a period of years.